# *Legionella pneumophila* exploits the endo-lysosomal network for phagosome biogenesis by co-opting SUMOylated Rab7

Chuang Li[1], Jiaqi Fu[1], Shuai Shao[2,3], Zhao-Qing Luo[1]*

**1** Purdue Institute of Inflammation, Immunology and Infectious Disease, Department of Biological Sciences, Purdue University, West Lafayette, Indiana, United States of America, **2** College of Pharmacy, The Ohio State University, Columbus, Ohio, United States of America, **3** Department of Biomedical Informatics, College of Medicine, The Ohio State University, Columbus, Ohio, United States of America

\* luoz@purdue.edu

**Data Availability Statement:** The authors confirm that the data supporting the findings of this study are in the manuscript and associated supporting information.

## Abstract

*Legionella pneumophila* strains harboring wild-type *rpsL* such as Lp02*rpsL*$_{WT}$ cannot replicate in mouse bone marrow-derived macrophages (BMDMs) due to induction of extensive lysosome damage and apoptosis. The bacterial factor directly responsible for inducing such cell death and the host factor involved in initiating the signaling cascade that leads to lysosome damage remain unknown. Similarly, host factors that may alleviate cell death induced by these bacterial strains have not yet been investigated. Using a genome-wide CRISPR/Cas9 screening, we identified *Hmg20a* and *Nol9* as host factors important for restricting strain Lp02*rpsL*$_{WT}$ in BMDMs. Depletion of *Hmg20a* protects macrophages from infection-induced lysosomal damage and apoptosis, allowing productive bacterial replication. The restriction imposed by *Hmg20a* was mediated by repressing the expression of several endo-lysosomal proteins, including the small GTPase Rab7. We found that SUMOylated Rab7 is recruited to the bacterial phagosome via SulF, a Dot/Icm effector that harbors a SUMO-interacting motif (SIM). Moreover, overexpression of Rab7 rescues intracellular growth of strain Lp02*rpsL*$_{WT}$ in BMDMs. Our results establish that *L. pneumophila* exploits the lysosomal network for the biogenesis of its phagosome in BMDMs.

## Author summary

The bacterial pathogen *Legionella pneumophila* extensively modulates vesicle trafficking pathways to create a membrane-bound vauole to support its intracellular growth. Although the role of the early phase of secretory pathway between the endoplasmic reticulum and the Golgi apparatus in the biogenesis of the Legionella-containing vacule (LCV) has been well-established, the role of the endosomal network in this process is poorly defined. Earlier studies using a *bona fide* wild-type *L. pneumophila* strain showed that Rab7, the small GTPase critical for the late endocytic pathway is enriched on the LCV but the mechanism of such recruitment is not understood. In this study, we used a genome-wide CRISPR/Cas9 screen to identify factors in mouse bone marrow-derived

**Funding:** National Institutes of Health grant R01AI127465 (ZQL). The funders had no role in study design, data collection and analysis, decision to publish, or preparation of the manuscript. Portions of salary of CL, FJ and ZQL were from the funders.

**Competing interests:** The authors have declared that no competing interests exist.

macrophages (BMDMs) that restrict intracellular growth of *L. pneumophila* strains such as Lp02$_{RpsLWT}$ harboring chromosomally encoded wild-type *rpsL* gene. We found that inactivation of *Hmg20a*, a gene involved in epigenetic regulation, allowed strain Lp02$_{RpsLWT}$ to grow in BMDMs. Further analysis revealed that *Hmg20a* indirectly governs this process by negatively controlling the expression of an array of genes involved in lysosomal biogenesis, including *rab7*. We found that overexpression of Rab7 alleviates the restriction of strain Lp02$_{RpsLWT}$ by BMDMs and only SUMOylated Rab7 can be recruited to the LCV. Finally, we identified the Dot/Icm substrate SulF as the bacterial factor responsible for Rab7 recruitment, which functions by binding to the SUMO moiety. Our results have established a molecular link between the LCV and the late endosome compartment, suggesting that *L. pneumophila* co-opts the endosomal network of host cells to support its intracellular growth.

## Introduction

One effective strategy of host immune defense in response to pathogen infection is induction of cell death [1]. Investigation of host responses triggered by infectious agents has led to the identification of pattern recognition receptors and novel immune mechanisms [2,3]. For instance, the observation that infection by *Shigella flexneri* induces macrophage death had paved the way to the discovery of pyroptotic cell death [4], which is fundamental for later identification of members of the Gasdermin family as executioner proteins of pyroptosis [5,6].

Study of the opportunistic bacterial pathogen *Legionella pneumophila* has revealed several unique features in bacterial pathogenesis, including the realization that a bacterial pathogen can dedicate more than 10% of its protein coding capacity to interact with its hosts [7,8] and that extremely diverse biochemical mechanisms are conferred by these effectors to modulate host cell signaling [9]. Examples include the autophagy inhibitor RavZ that directly cleaves ATG8 proteins attached to phosphatidylethanolamine on autophagosome membranes [10], and members of the SidE family that induce phosphoribosyl ubiquitination by a mechanism that is chemically distinct from conventional ubiquitination catalyzed by the E1-, E2- and E3-enzyme cascade [11,12,13]. Equally exciting is the discovery of several new insights in host immunity by using this model [14]. For example, investigation of the responses of different mouse lines to *L. pneumophila* [15] led to the discovery of NAIP5 as the sensor for flagellin involved in the activation of the NLRC4 inflammasome [16,17]; the analysis of effectors that function to inhibit protein synthesis provided the clues that cells can mount a specific immune induction in response to protein synthesis inhibition [18,19] and the finding that infected macrophages enable alveolar epithelium to produce granulocyto-macrophage colony-stimulating factor (GM-CSF) that induces antibacterial inflammation by metabolic reprogramming [20].

In addition to its established role as the cellular "recycling bins", the lysosome has been increasingly recognized as a signal integration hub for numerous cellular processes, ranging from calcium regulation in nutrient response to immunity [21]. Dysregulation in lysosomes has been linked to a variety of diseases, including tumor infiltration[22] and viral pathogenesis [23]. Lysosomal membrane permeabilization (LMP), a cellular process triggered by many forms of cell stress, is characterized by the translocation of lysosomal enzymes from the lumen into the cytoplasm [24]. Although the precise molecular mechanisms governing the regulation of lysosome protease release and the maintenance of lysosome membrane integrity have yet to be fully elucidated, the extent of proteases leakage determines the outcomes of the stress, ranging from autophagy to cell death [25].

Our previous study found that *L. pneumophila* strains harboring chromosomal wild-type *rpsL* gene induce LMP in mouse BMDMs, resulting in death of infected cells and termination of bacterial infection [26]. RpsL is the S12 component of the bacterial ribosome and is the target of antibiotics such as streptomycin that function by inhibiting protein synthesis. Most spontaneous mutations that confer resistance to streptomycin are mapped to K88R or K43N in the *rpsL* gene, these mutations reduced the binding affinity beween RpsL and the antibiotic by 2–3 magnitude [27,28]. Strikingly, although the MIC of the K43N and K88R mutants to streptomycin is indistinguishable, only the K88R mutation licensed *L. pneumophila* strains to grow in mouse bone marrow-derived macrophages (BMDMs) [26]. However, the mechanism by which *L. pneumophila* strains expressing chromosomal wild-type RpsL triggers lysosomal membrane damage and the host processes implicated in restricting bacterial replication under this condition remain inadequately understood. We have postulated that *L. pneumophila* strains harboring chromosomal wild-type *rpsL* release specific ligands that are recognized by BMDM-specific receptors, thereby initiating a signaling cascade to trigger lysosomal damages and macrophage death [29]. Inactivation of the genes coding for the putative receptors or components of this signaling pathway in BMDMs is expected to ablate macrophage death, hence allowing productive intracellular bacterial proliferation. Using genome-wide CRISPR/Cas9 knockout screenings [30,31], we here identified *Hmg20a* and *Nol9* as host factors involved in restricting intracellular replication of the strain Lp02*rpsL*$_{WT}$ in BMDMs. Among these, Hmg20a functions by regulating the abundance of lysosomal proteins, such as Rab7, overexpression of which endowed mouse BMDMs to support its intracellular replication. Furthermore, Rab7 was enriched on phagosomes harboring strain Lp02*rpsL*$_{WT}$ by a mechanism that requires SUMOylation and the Dot/Icm effector SulF, which interacts with SUMOylated Rab7 by a SUMO-interacting motif (SIM) to inhibit its GTPase activity by preventing the binding of a GTPase activation protein (GAP).

## Results

### Identification of *Hmg20a* and *Nol9* as factors involved in restricting intracellular replication of the *L. pneumophila* strain Lp02*rpsL*$_{WT}$

Genome-wide CRISPR/Cas9 screening utilizing sgRNA libraries targeting all genes in the mouse genome is a powerful tool to identify factors important for specific biological processes [32,33,34,35]. Here we adapted this approach to search for host factors involved in limiting intracellular replication of *L. pneumophila* (**S1A Fig**). To this end, we first crossed a *Cas9*$^{+/+}$ mouse line [5] with A/J mice to obtain Cas9-expressing mice permissive for *L. pneumophila* infection [36] (**S1B Fig**). Next, we challenged BMDMs from this mouse line with relevant *L. pneumophila* strains to evaluate bacterial growth by counting total colony formation units (CFUs) and by assessing the formation of large replicative vacuoles. Our results showed that whereas the common laboratory strain Lp02*rpsL*$_{K88R}$ [37] grew robustly in BMDMs isolated from *Cas9*$^{+/+}$ A/J mice, strain Lp02*rpsL*$_{WT}$ or Lp02*rpsL*$_{K43N}$ [26] cannot replicate [38], indicating that BMDMs from this mouse line are suitable for CRISPR/Cas9-based screenings (**Fig 1A–1C**).

We performed the screening by transducing BMDMs prepared from *Cas9*$^{+/+}$ A/J mice with lentiviral particles carrying the sgRNA library [39], followed by infections with strain Lp02*rpsL*$_{WT}$(pAM239) (GFP$^+$) (**Fig 1D**). At 14 h post-bacterial internalization, infected macrophages potentially harboring large replicative vacuoles were collected by fluorescence-activated cell sorting (FACS) (**Figs 1D and S2**). To ensure the enrichment of cells containing large Legionella-containing vacuoles (LCVs), pulse-shape analysis (PulSA) which recognizes cytoplasmic large protein aggregates was applied to the flow cytometry procedure [40]

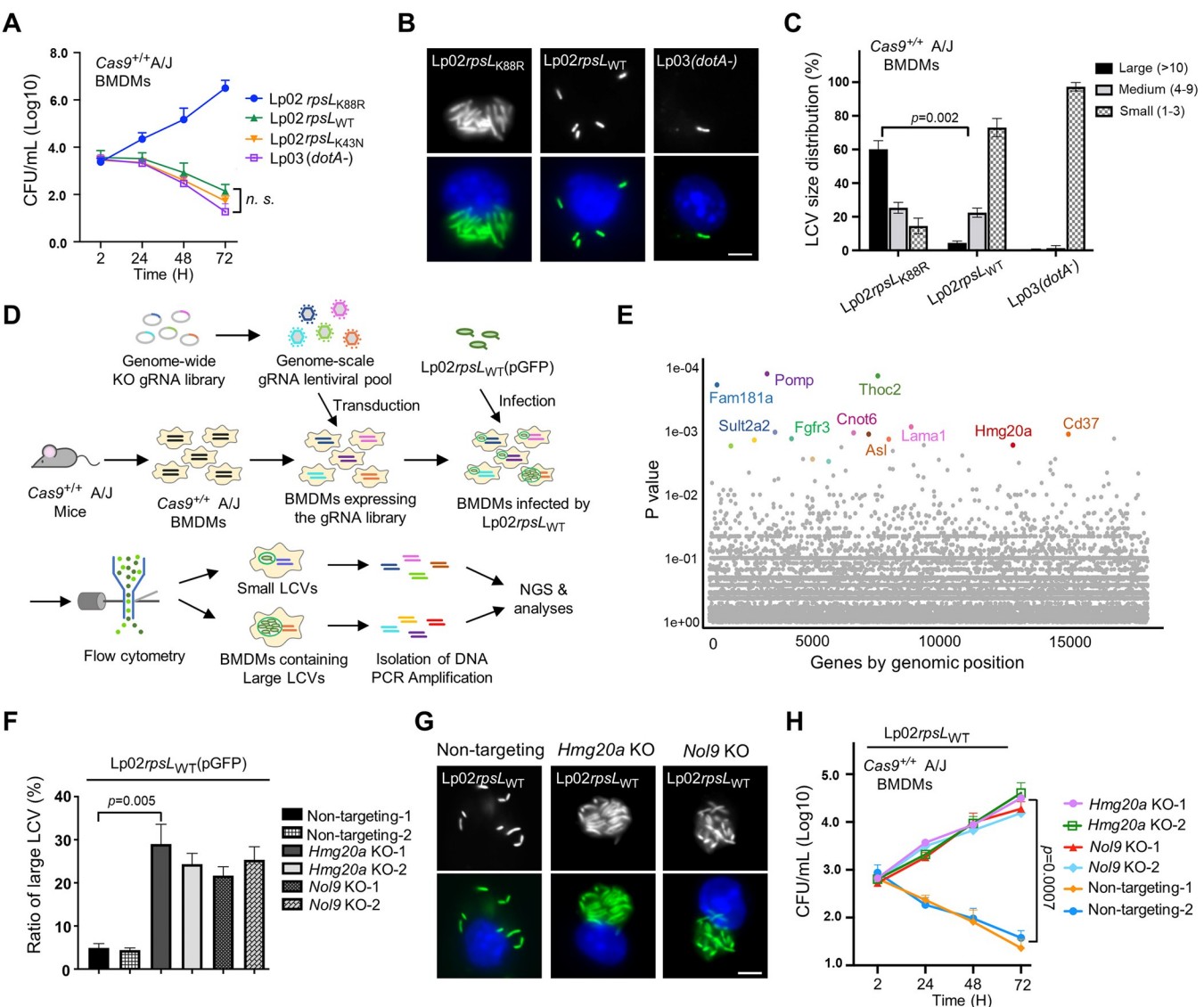

**Fig 1. Identification of host factors involved in restricting intracellular growth of strain Lp02rpsL_WT in BMDMs. (A)** Intracellular growth of relevant *L. penumophila* strains in BMDMs from $Cas9^{+/+}$ A/J mice. The indicated *L. penumophila* strains were used to infect BMDMs at an MOI of 0.05 and bacterial growth was determined by plating lysates of infected cells at the indicated time points. **(B-C)** The formation and size distribution of bacterial phagosome. BMDMs from $Cas9^{+/+}$ A/J mice were infected with relevant *L. pneumophila* strains for 14 h and samples were processed for microscopic analysis for acquiring representative images of large LCVs **(B)** or their size distribution **(C)**. Bar, 5 μm. LCVs containing more than 10 bacteria were defined as large; those containing 4 to 9 bacteria were classified as medium and those containing 1 to 3 bacteria were categorized as small vacuoles. **(D)** Schematic workflow of the genome-scale CRISPR/Cas9 screening. BMDMs from $Cas9^{+/+}$ A/J mice were transduced with lentiviral particles carrying the CRISPR knockout sgRNA library for 6 d, cells were then infected with strain Lp02rpsL_WT(pGFP) for 14 h. Fixed samples were used to isolate cells harboring large LCVs by flow cytometry and DNA was extracted and processed for analysis by next generation sequencing (NGS). **(E)** Gene enriched in BMDMs containing large LCVs vs those containing small LCVs shown by Dot plot. The significance of distribution of genes was determined based on the statistical significance (y axis) versus genomic position (x axis). The top 10 genes were highlighted. **(F-H)** Knockout of *Hmg20* or *Nol9* allowed strain Lp02rpsL_WT to replicate in BMDMs. BMDMs were transduced by lentivirus particles carrying sgRNAs against *Hmg20* or *Nol9* and cells were infected with *L. pneumophila* for 14 h to assess intracellular replication by determining the distribution of bacterial phagosome in size **(F and G)** and total bacterial counts over 72 h at 24h intervals **(H)**. In each case, for quantitative experiments, results (mean ± s.e.) were from three independent experiments each done in triplicate. Images shown were representatives from one experiment.

(**S2A–S2E Fig**). As a control, cells that exhibited minimal GFP signals were also collected (**S2D–S2E Fig**). To determine the effectiveness of the cell sorting in enriching cells harboring large LCVs, we stained a fraction of the samples and confirmed the presence of large LCVs in these cells by microscopic analysis. Approximately 63% of the cells exhibiting strong

fluorescence signals harbored vacuoles of more than 10 bacteria (P4 sample, **S2F and S2G Fig**). In contrast, samples collected by the property of weak GFP signals did not contain any large vacuole (P5 sample, **S2F and S2G Fig**). Next, we determined the sequences and abundances of sgRNAs in infected cells harboring large LCVs by next generation sequencing (NGS) [41] following PCR amplification using an established method [5]. After volcano plot of confidence score (casTLE score) analyses, candidate genes with comparatively higher fold changes in sgRNA reads were chosen for further investigation [35] (**Figs 1E** and **S1C**).

Out of the approximately 17,500 genes identified by sequencing analyses, we selected the leading sixty candidates based on their fold changes in samples containing large LCVs compared to controls. The importance of these genes in restricting intracellular growth of strain Lp02$rpsL_{WT}$ in BMDMs was individually examined by gene knockout. Whereas sgRNAs targeting most of these genes did not yield robust intracellular bacterial growth, those targeting *Hmg20a* or *Nol9* led to approximately five-fold higher intracellular replication than the controls (**Fig 1F–1H**). Furthermore, strain Lp02$rpsL_{WT}$ formed large replicative vacuoles in macrophages lacking *Hmg20a* (**Fig 1G**). Thus, we further investigated the mechanism by which *Hmg20a* regulates intracellular replication of strain Lp02$rpsL_{WT}$ in BMDMs.

## Deletion of *Hmg20a* allows growth of Lp02$rpsL_{WT}$ in mouse BMDMs

To verify the intercellular growth phenotypes, we purchased a *Hmg20a$^{+/-}$* mouse line and attempted to obtain homozygous *Hmg20a$^{-/-}$* mice. Because homozygous *Hmg20a$^{-/-}$* mice are embryonically lethal, we generated an *Hmg20a$^{+/-}$ Cas9$^{+/+}$* mouse line in the A/J background by mating heterozygous *Hmg20a$^{+/-}$* mice with *Cas9$^{+/+}$* A/J mice over four generations (**S3A Fig**). BMDMs from these mice supported intracellular growth of the common laboratory strain Lp02$rpsL_{K88R}$ but not Lp02$rpsL_{WT}$ (**S3B and S3C Fig**).

Next, we transduced these BMDMs with lentiviral particles carrying sgRNAs targeting two distinct regions of the *Hmg20a* gene using a procedure that permits efficient gene knockout in primary macrophages [42] (**Fig 2A**). Cells transduced with viral particles carrying a non-targeting sgRNAs were established as controls. In each case, non-transduced cells were eliminated by puromycin treatment for 2 d and surviving cells were collected to evaluate the efficiency of *Hmg20a* knockout (**Fig 2A**). Immunoblot assessment using Hmg20a-specific antibodies revealed that the protein level of Hmg20a was reduced by approximately 90% in cells receiving the sgRNAs targeting this gene (**Fig 2B and 2C**).

We next evaluated the replication of *L. pneumophila* strains in *Hmg20a$^{-/-}$* macrophages. In BMDMs deficient in *Hmg20a*, robust replication of strain Lp02$rpsL_{WT}$ was observed, whereas Hmg20a-positive cells failed to support bacterial replication (**Fig 2D**). Approximately 30% of the LCVs harbored more than 10 bacteria in BMDMs in which Hmg20a had been depleted. The ratio of such large LCVs in cells expressing Hmg20a was less than 5% (**Fig 2E**). We also assessed the replication by determining total bacterial counts. Only BMDMs lacking *Hmg20a* supported robust growth of Lp02$rpsL_{WT}$ for the entire 72 h experimental duration (**Fig 2F**). Taken together, these results establish that *Hmg20a* negatively impacts the growth of Lp02$rpsL_{WT}$ in mouse BMDMs.

## *Hmg20a* knockout protects lysosomal integrity in macrophages infected by strain Lp02$rpsL_{WT}$

One feature associated with the infection of strain Lp02$rpsL_{WT}$ in BMDMs is lysosomal membrane permeabilization (LMP) [26], which is known to cause the release of cathepsins from the lysosomal lumen, leading to the cleavage of Bid to generate tBid [43]. tBid then triggers the

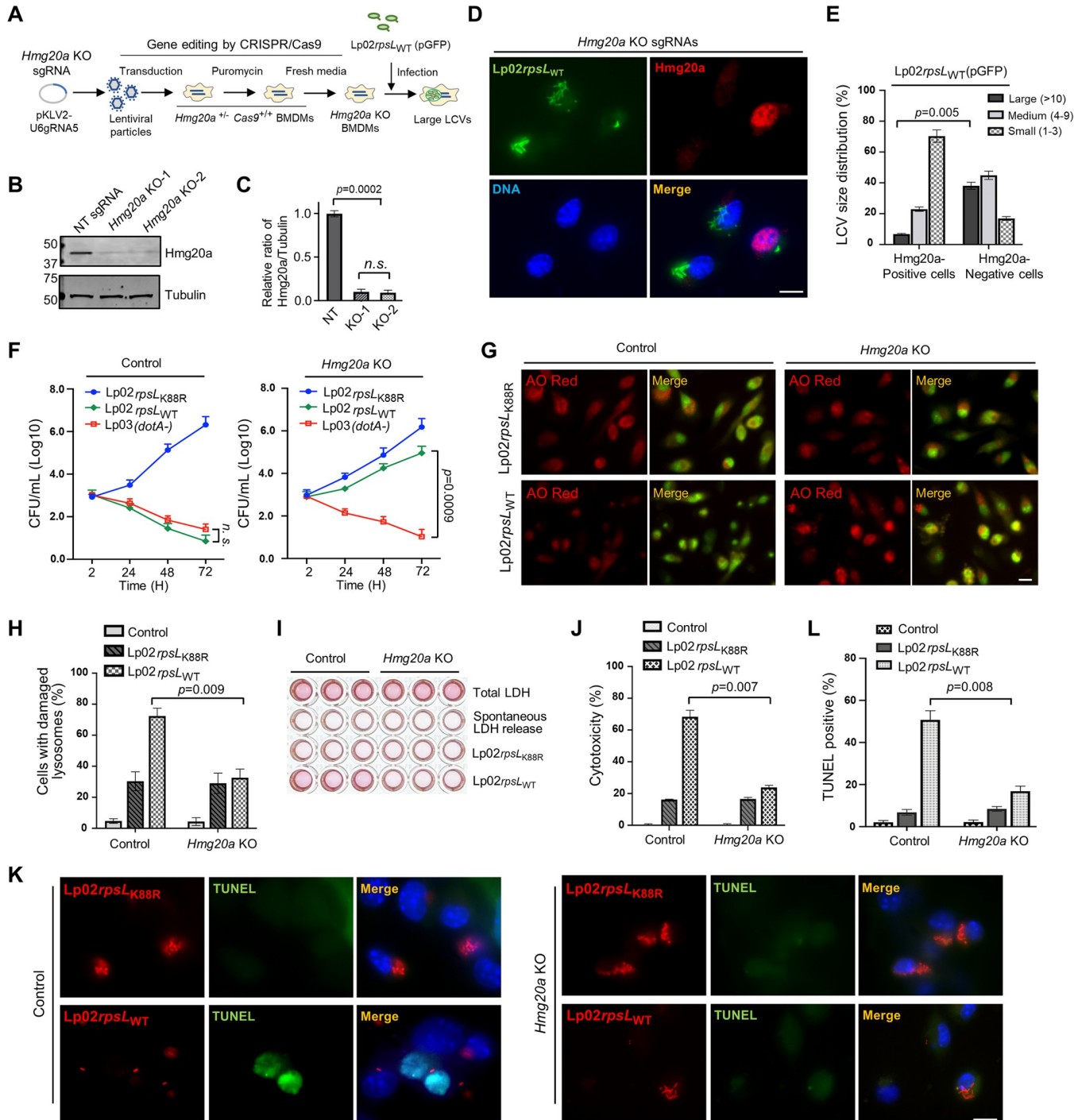

**Fig 2. Deletion of *Hmg20a* allows strain Lp02*rpsL*_WT to grow in mouse BMDMs. (A)** Schematic of CRISPR/Cas9-mediated *Hmg20a* knockout in BMDMs and infection with strain Lp02*rpsL*_WT(pGFP). BMDMs from *Hmg20*^+/− *Cas9*^+/+ A/J mice were transduced with lentiviruses carrying sgRNAs targeting *Hmg20a* for 3 d, non-transduced cells were eliminated by incubation in medium containing puromycin for 2 d. BMDMs seeded in medium without puromycin for an additional 3 d were infected with the bacteria for 14 h. **(B-C)** *Hmg20a* knockout efficiency in BMDMs. Lysates of cells treated as described in A were subjected to immunoblotting using anti-Hmg20a antibodies **(B)** and the rates of reduction were evaluated by measuring band density relative to bands of the loading control tubulin from three independent experiments **(C)**. **(D-F)** Growth of strain Lp02*rpsL*_WT in *Hmg20a*-deleted BMDMs. BMDMs treated as described in A were infected with strain Lp02*rpsL*_WT at an MOI of 1 for 14 h and samples were processed for image acquisition **(D)**, determining the distribution of the phagosome in size **(E)**. Bar, 10 μm. Similar experiments with an MOI of 0.05 were used to analyze intracellular replication by determining total bacterial counts at the indicated time points **(F)**. **(G-L)** *Hmg20a* knockout protected BMDMs from infection-induced lysosome damage and cell death. BMDMs received sgRNAs targeting *Hmg20a* or nontarget were infected with strain Lp02*rpsL*_WT or Lp02*rpsL*_K88R, samples loaded with AO were processed for image acquisition

(G), for determining the ratios of cells with damaged lysosomes by AO staining (H). Cell death was measured by the release of lactate dehydrogenase (LDH) (I and J) and by TUNEL staining (K and L). In each case, quantitative results (mean ± s.e.) were from three independent experiments each done in triplicate. Images are representatives of one of the three experiments with similar results. Bar, 10 μm.

release of cytochrome c (Cyto c) from mitochondria, which activates the classic apoptotic pathway that ultimately leads to caspase-3 activation and cell death [44].

We set out to determine the impact of *Hmg20a* depletion on lysosome integrity in BMDMs infected with Lp02$rpsL_{WT}$ via acridine orange (AO) straining and the release of lactate dehydrogenase (LDH). AO is a fluorescent dye that, when concentrated in the lysosome, emits orange fluorescence upon blue light excitation, but redistributes itself into the cytosol and nuclei in cells with damaged lysosome membranes and displays green fluorescence signals [45]. Infections with strain Lp02$rpsL_{WT}$ in *Hmg20a*$^{-/-}$ BMDMs caused lysosome damage in about 30% cells, which was similar to infections with strain Lp02$rpsL_{K88R}$ on wild-type and *Hmg20a*$^{-/-}$ BMDMs (**Fig 2G and 2H**). In contrast, this strain induced such damage in approximately 72% BMDMs from wild-type mice (**Fig 2G and 2H**). Similarly, when infected with Lp02$rpsL_{WT}$, the level of LDH release in *Hmg20a* knockout cells decreased to approximately 20% from 67% of BMDMs treated with nontargeting sgRNAs (**Fig 2I and 2J**).

## Knockout of *Hmg20a* protects macrophages from cell death induced by strain Lp02$rpsL_{WT}$ infection

To corroborate the effects of *Hmg20a* on lysosome integrity, we analyzed events downstream of LMP in *Hmg20a*-deficient BMDMs infected with *L. pneumophila*. One hallmark of apoptosis is extensive DNA fragmentation, which can be detected by the TdT-mediated dUTP-biotin nick end labeling (TUNEL) assay [46]. Similar to results from previous studies [26], infection of wild-type BMDMs with strain Lp02$rpsL_{WT}$ elicited extensive cell death, with nearly 40% of the infected cells being TUNEL positive (**Fig 2K and 2L**). Yet, in *Hmg20a* knockout BMDMs, the rate of cell death was only about 10% (**Fig 2K and 2L**). In line with the decreased cell death rates, infection with strain Lp02$rpsL_{WT}$ caused less cleavage of caspase-3 and PARP in *Hmg20a* knockout BMDMs than in wild-type cells (**S3D and S3E Fig**). Taken together, these results establish that depletion of *Hmg20a* stabilized lysosomes in Lp02$rpsL_{WT}$-infected macrophages.

## Hmg20a impacts the abundance of proteins involved in lysosomal function

Hmg20a is a nuclear protein homologous to Hmg20b which participates in the assembly of the Lys-specific demethylase 1/REST co-repressor 1 (LSD1-CoREST) histone demethylase complex involved in epigenetic regulation of gene expression [47]. Therefore, it is highly probable that Hmg20a indirectly impacts the cellular processes that regulate the integrity of lysosome membranes in response to Lp02$rpsL_{WT}$ infection.

To identify proteins subjected to regulation by Hmg20a, we conducted comparative proteomic analyses of *Hmg20a*$^{-/-}$ and wild-type BMDMs. Whereas the abundance of the majority of proteins detected was similar between these two samples, the level of a number of proteins involved in lysosome biogenesis, including Rab7 and the lysosomal-associated membrane protein 1 (LAMP1) were elevated in *Hmg20a* knockout cells (**Fig 3A and 3B**). Similar variations were observed in a previous study which comparatively analyzed proteomics of cells in which *Hmg20a* was knocked down by siRNA [48]. We validated the proteomic results by probing Rab7 and LAMP1 by immunoblotting and immunofluorescence staining, which revealed that knocking out of *Hmg20a* led to 2–3 folds increase in the abundance of Rab7 and LAMP1 in BMDMs, respectively (**Fig 3C and 3D**). In contrast, this manipulation had no discernible effect

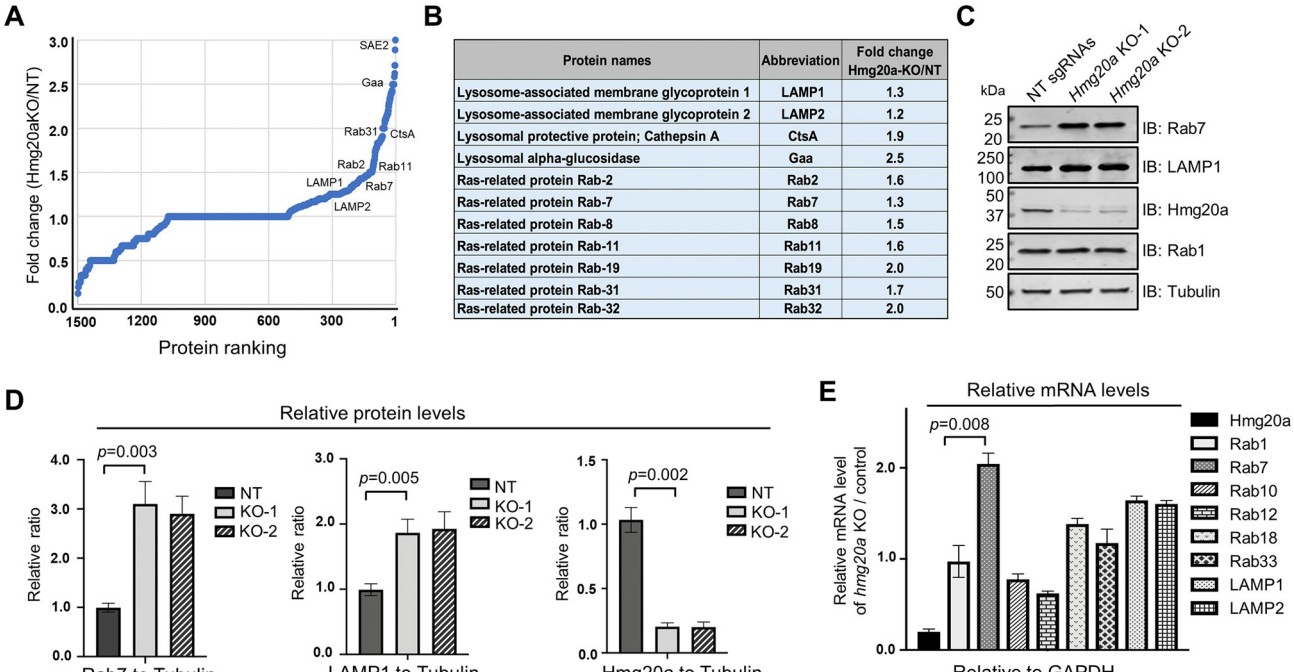

**Fig 3. Hmg20a impacts the abundancy of proteins involved in lysosome function. (A)** The protein abundancy of *Hmg20a⁺ᐟ⁻* and *Hmg20a⁻ᐟ⁻* BMDMs was analyzed by mass spectrometry and the comparative proteomic results were shown by a dot plot graph. Several representative proteins with marked changes were labeled in the graph. **(B)** Fold change of representative lysosome-related proteins in *Hmg20a⁻ᐟ⁻* cells relative to *Hmg20⁺ᐟ⁻* cells. **(C-D)** Evaluation of the change in protein abundancy by Immunoblotting **(C)**, the changes were quantitated by measuring band intensity with tubulin as the reference **(D)**. Results shown (mean ± s.e.) were from three independent experiments. **(E)** Comparison of mRNA levels of several Rab proteins between *Hmg20a⁺ᐟ⁻* and *Hmg20a⁻ᐟ⁻* BMDMs. Total mRNA was extracted and analyzed via RT-qPCR using primers specific for *Hmg20a*, *Rab1*, *Rab7*, *Rab10*, *Rab12*, *Rab18*, *Rab33*, *LAMP1* or *LAMP2*. *GAPDH* was used as the control gene.

on the level of Rab1, which is mostly localized at the endoplasmic reticulum (ER) and ER-Golgi intermediate compartment (ERGIC) [49] (**Fig 3C and 3D**). In agreement with the putative role of *Hmg20a* in epigenetic regulation of gene expression, knockout of *Hmg20a* increased the mRNA levels of Rab7 and LAMP1/2 in mouse BMDMs by 1.5–2 folds but had no impact on the transcription of Rab1 or Rab10 (**Fig 3E**).

## Overexpression of Rab7 permits productive intracellular replication of strain Lp02*rpsL*$_{WT}$ in BMDMs

Next, we examined whether any of these Rab proteins was implicated in intracellular replication of strain Lp02*rpsL*$_{WT}$ in BMDMs by gene knockout or overexpression. Whereas knockout of *Rab1*, *Rab7*, *Rab12*, *Rab18* or *Rab33* did not detectably impact intracellular growth of strain Lp02*rpsL*$_{WT}$ in BMDMs (**Fig 4A**), ectopic expression of Rab7 but not Rab12 significantly increased the ratio of large LCVs formed by strain Lp02*rpsL*$_{WT}$ (**Fig 4B and 4C**). Thus, the rescue of Lp02*rpsL*$_{WT}$ growth in BMDMs was Rab7-specific. Similar results were obtained from experiments in which bacterial growth was measured by determining total CFUs (**Fig 4D**). Furthermore, the ratio of cell death of BMDMs transduced to overexpresss Rab7 induced by infection with strain Lp02*rpsL*$_{WT}$ was significantly lower than that of similarly infected normal cells (**S4A–S4C Fig**). However, overexpression of Rab7 in BMDMs did not detectablly increase intracellular growth of strain Lp02*rpsL*$_{K88R}$, which grows robustly in normal BMDMs as shown in numerous studies (**S4D–S4F Fig**). We also observed that the growth of strain Lp02*rpsL*$_{WT}$ in BMDMs overexpressing Rab7 mostly occurred in the first 24 h post infection

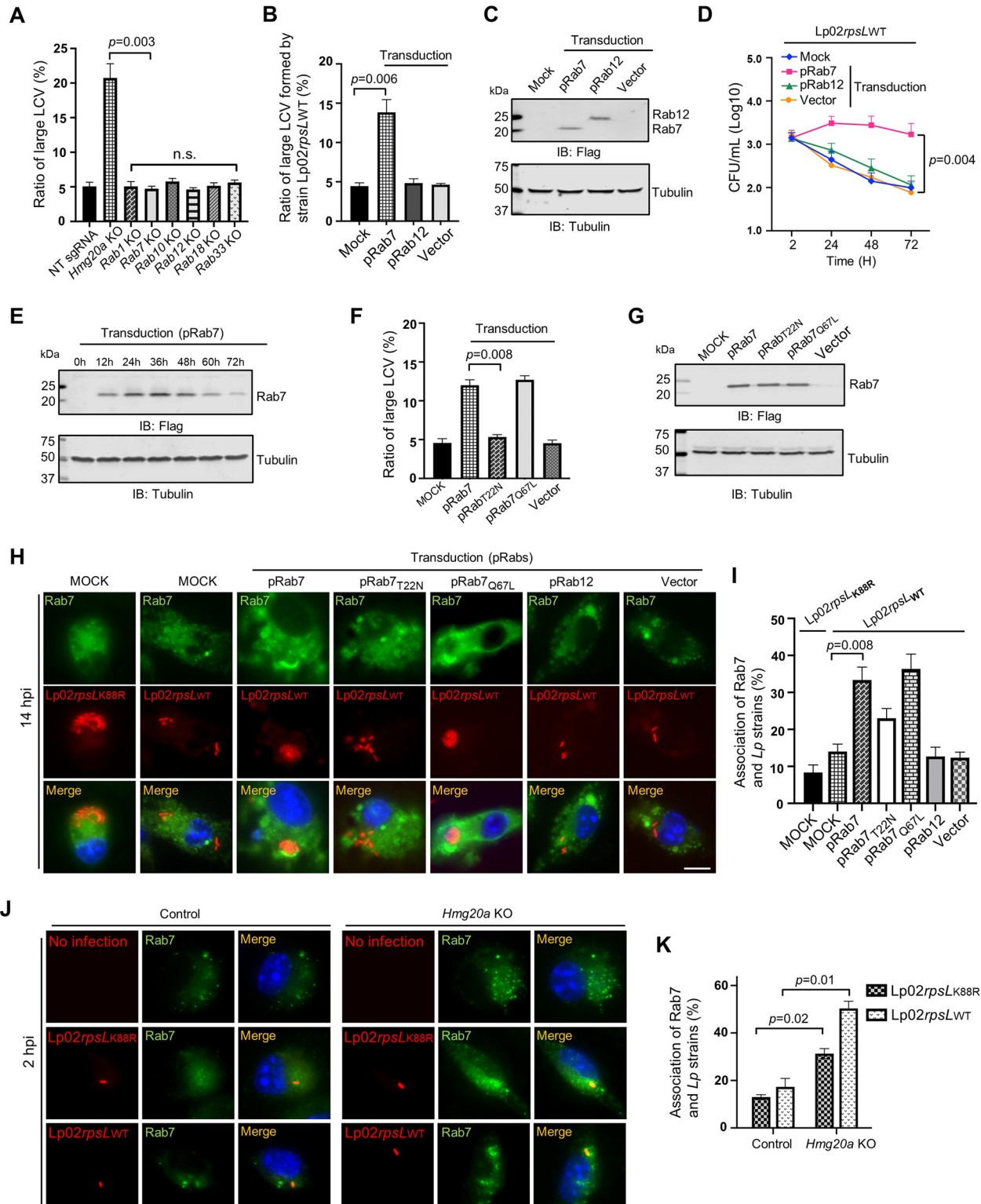

**Fig 4. Overexpression of Rab7 permits productive intracellular replication of strain Lp02*rpsL*$_{WT}$ in BMDMs. (A)** Knockout of *Hmg20a* but not several Rab genes allowed strain Lp02*rpsL*$_{WT}$ to grow in BMDMs. SgRNAs targeting *Rab1*, *Rab7*, *Rab10*, *Rab12*, *Rab18*, *Rab33* or *Hmg20a* were introduced into BMDMs and the cells were infected with *L. pneumophila* at an MOI of 1 for 14 h prior to being processed for analysis of the formation of large LCVs. **(B-E)** Overexpression of Rab7 promotes growth of strain Lp02*rpsL*$_{WT}$ in BMDMs. Cells transduced to express Flag-Rab7 were used to determine bacterial growth by the formation and distribution of LCVs in size **(B)** and total bacterial counts in a 72-h experimental

duration (**D**). The abundancy of ectopically expressed Flag-Rab7 was determined by immunoblotting with the Flag antibody (**C and E**). Note that Lp02*rpsL*$_{WT}$ infection started at 24 h post lentiviral transduction. (**F-G**) Only active Rab7 suppressed the restriction of the growth of strain Lp02*rpsL*$_{WT}$ by BMDMs. Cells transduced to express Flag-Rab7 or its mutants were infected with strain Lp02*rpsL*$_{WT}$ at an MOI of 1 for 14 h, samples were processed for analyzing the formation and distribution of LCVs (**F**). The expression of Rab7 and its mutant was detected by immunoblotting with the Flag antibody (**G**). (**H-I**) The association of Rab7 and its mutants with LCVs formed by Lp02*rpsL*$_{WT}$ in BMDMs. Cells transfected to express Flag-Rab7 or its mutants were infected with the common laboratory *L. pneumophila* strain Lp02*rpsL*$_{K88R}$ or strain Lp02*rpsL*$_{WT}$ for 14 h prior to immunostaining with antibodies specific for Rab7. Images were acquired using an Olympus X-81 microscope (**H**) and the distribution of LCVs was determined by counting at least 300 infected cells (**I**). (**J-K**) Rab7 recruitment by strains Lp02*rpsL*$_{K88R}$ and Lp02*rpsL*$_{WT}$. *Hmg20a*$^{+/-}$ or *Hmg20a*$^{-/-}$ BMDMs were infected with the bacteria for 2 h and the recruitment of Rab7 to LCVs was analyzed by immunostaining (**J**) and the ratios of such association were determined (**K**). In all cases, quantitative results (mean ± s.e.) were from three independent experiments each done in triplicate. Images shown were a representative from three independent experiments with similar results. Bar, 10 μm.

(**Fig 4D**), which suggested that the elevation of Rab7 by transduction may not persist throughout the entire infection duration. Thus, we examined the kinetics of Rab7 in transduced BMDMs and found that its protein level began to decline after reaching the peak at 36 h post transduction (**Fig 4E**). These data further indicate that the gain of growth of strain Lp02*rpsL*$_{WT}$ in these cells in the first 24 h was a result of elevated Rab7 protein level.

## Active Rab7 is required to support the replication of strain Lp02*rpsL*$_{WT}$ in BMDMs

Rab7 is a small GTPase that participates in vesicle trafficking between late endocytic compartments, including late endosomes and lysosomes [50,51]. It is critical for lysosomal biogenesis, localization, functions and signaling [52]. The GTP-bound form of Rab7 is active in signaling, which becomes the GDP-bound inactive form after GTP is hydrolyzed by its GTPase activity, a process that is stimulated by GTPase activation factors [51]. To examine which form of Rab7 is capable of rescuing the growth of strain Lp02*rpsL*$_{WT}$ in BMDMs, we expressed Rab7$_{Q67L}$ and Rab7$_{T22N}$ which mimics the GTP-bounded active form and the GDP-bound inactive form of the small GTPase, respectively [53] in BMDMs and assessed the replication of this strain. Our results indicated that overexpression of Rab7$_{Q67L}$ but not Rab7$_{T22N}$ increased the ratio of large LCVs formed by strain Lp02*rpsL*$_{WT}$ in BMDMs (**Fig 4F and 4G**). Thus, Rab7 in its GTP-bound active status supports the replication of strain Lp02*rpsL*$_{WT}$ in BMDMs, likely by participating in one or more signaling events.

## Rab7 is enriched on *Legionella* phagosomes

Rab7 is known to be associated with the lysosome-like phagosome of *Coxiella burnetii* [54] as well as the biogenesis of vacuoles that support the growth of *Salmonella typhimurium* [55,56] or *Helicobacter pylori* [57]. Yet, its role in *L. pneumophila* infection, particularly in the scenario of strains such as Lp02*rpsL*$_{WT}$ is unknown. We thus examined the distribution of Rab7 in cells infected with relevant *L. pneumophila* strains. Ectopically expressed Rab7 but not its Rab7$_{T22N}$ mutant was enrichment on LCVs formed by strain Lp02*rpsL*$_{WT}$ in BMDMs (**Fig 4H and 4I**). Under the same conditions, enrichment of Rab7 on LCVs formed by strain Lp02*rpsL*$_{K88R}$ also occurred, but the signals were considerably weaker than those seen with Lp02*rpsL*$_{WT}$ (**Fig 4H and 4I**). Furthermore, knockout of *Hmg20a* in BMDMs caused an increase in Rab7 abundancy and more pronounced enrichment on LCVs formed by strain Lp02*rpsL*$_{WT}$ (**Fig 4J and 4K**).

We also examined the association of several proteins of different organelles, including Rab1 (ER/Golgi), LAMP1 (lysosome), EEA1 (early endosome), Calnexin (ER) and GM130 (the Golgi apparatus) with LCVs containing strain Lp02*rpsL*$_{WT}$ in BMDMs overexpressing Rab7, which revealed that proteins associated with the lysosome, but not the Golgi apparatus, were enriched on LCVs containing this particular strain (**S5A–S5D Fig**). In addition, ER proteins

such as Rab1 and Calnexin were also associated with LCVs of strain Lp02*rpsL*$_{WT}$ (**S5A–S5D Fig**). We further dissected the differences of vacuoles of strains Lp02*rpsL*$_{WT}$ and Lp02*rpsL*$_{K88R}$ in recruiting Rab7 by co-infecting BMDMs with these two strains. While Rab1 was recruited by both strains, Rab7 was preferably enriched on vacuoles formed by strain Lp02*rpsL*$_{WT}$ (**S5E and S5F Fig**). Collectively, these data demonstrate that proteins involved in lysosome biogenesis are more robustly recruited to the membrane of LCVs harboring strain Lp02*rpsL*$_{WT}$ in BMDMs.

## SUMOylation is required for the recruitment of Rab7 by LCVs

To elucidate the mechanism of Rab7 recruitment by LCVs, we sought to identify interacting *L. pneumophila* proteins. Affinity pulldown experiments with recombinant Rab7 purified from *Escherichia coli* did not yield any meaningful hit. Rab7 is known to undergo several post-translational modifications (PTMs), including ubiquitination, phosphorylation, palmitoylation and SUMOylation [58] (**Fig 5A**), we thus questioned whether any of these modifications contributes to its recruitment to LCVs. To this end, we created substitution mutants in Rab7 sites known to receive these PTMs and analyzed their ability to be recruited to LCVs formed by strain Lp02*rpsL*$_{WT}$. Whereas mutations in K38, S72 and C83/84 of Rab7 responsible for receiving ubiquitination, phosphorylation and palmitoylation respectively, did not significantly affect its recruitment to LCVs, a substitution in K175 involved in SUMOylation [59] drastically decreased such recruitment (**Fig 5B and 5C**). In line with these results, ectopic expression of the SUMOylation-deficient Rab7 failed to rescue the replication of strain Lp02*rpsL*$_{WT}$ in BMDMs (**Fig 5D–5F**). Thus, Rab7 SUMOylation is important for its recruitment to the LCV.

## The Dot/Icm substrate SulF recruits SUMOylated Rab7 to LCVs via a SUMO- interacting motif

The recruitment of small GTPases to the phagosome of *L. pneumophila* often is mediated by its Dot/Icm effectors [60]. In light of the finding that SUMOylation is important for Rab7 recruitment, we analyzed the presence of potential SUMO-interacting motif (SIM) in known Dot/Icm substrates [61] using the group-based prediction system GPS-SUMO/SIM [62]. These efforts retrieved nine effectors with a putative SIM motif. Among these, the hydrophobic core of the predicted SIM in Lpg2832 was identical to that found in the SUMO-targeted ubiquitin E3 ligase RNF4 [63] (**Fig 5G**). We first evaluated their binding to SUMO by incubating GST-tagged recombinant protein of each effector with His$_6$-SUMO, which revealed that six of these effectors, including Lpg0376, Lpg0519, Lpg0963, Lpg1798, Lpg2832 and Lpg2912, detectably interacted with SUMO (**S6A and S6B Fig**). The ability of these proteins to interact with SUMOylated Rab7 was examined by immunoprecipitation, and only Lpg2832 was found to reproducibly bind SUMO-Rab7 (**S6C Fig**). Furthermore, mutations in the SIM hydrophobic core of Lpg2832 abolished its interactions with SUMO and SUMOylated Rab7 (**Fig 5H and 5I**). Because of these features, we designated Lpg2832 as <u>SU</u>MO-interacting <u>L</u>egionella <u>f</u>actor (SulF).

To examine whether SulF is sufficient for Rab7 recruitment, we deleted this gene from both strains Lp02*rpsL*$_{WT}$ and Lp02*rpsL*$_{K88R}$ (**S6D Fig**). Loss of *sulF* ablated the enrichment of Rab7 on LCVs formed by both strains in BMDMs ectopically expressing Rab7 (**Fig 5J and 5K**). Furthermore, Lp02*rpsL*$_{WT}$Δ*sulF* displayed growth defects in these cells (**Fig 5L and 5M**). The defects in Rab7 enrichment and intracellular growth exhibited by strain Lp02*rpsL*$_{WT}$Δ*sulF* can be restored by expressing SulF but not SulFIV/AA defective in binding SUMOylated Rab7 (**Fig 5J–5N**). Deleting *sulF* in strain Lp02*rpsL*$_{K88R}$ did not impair its intracellular replication in BMDMs (**S6E and S6F Fig**).

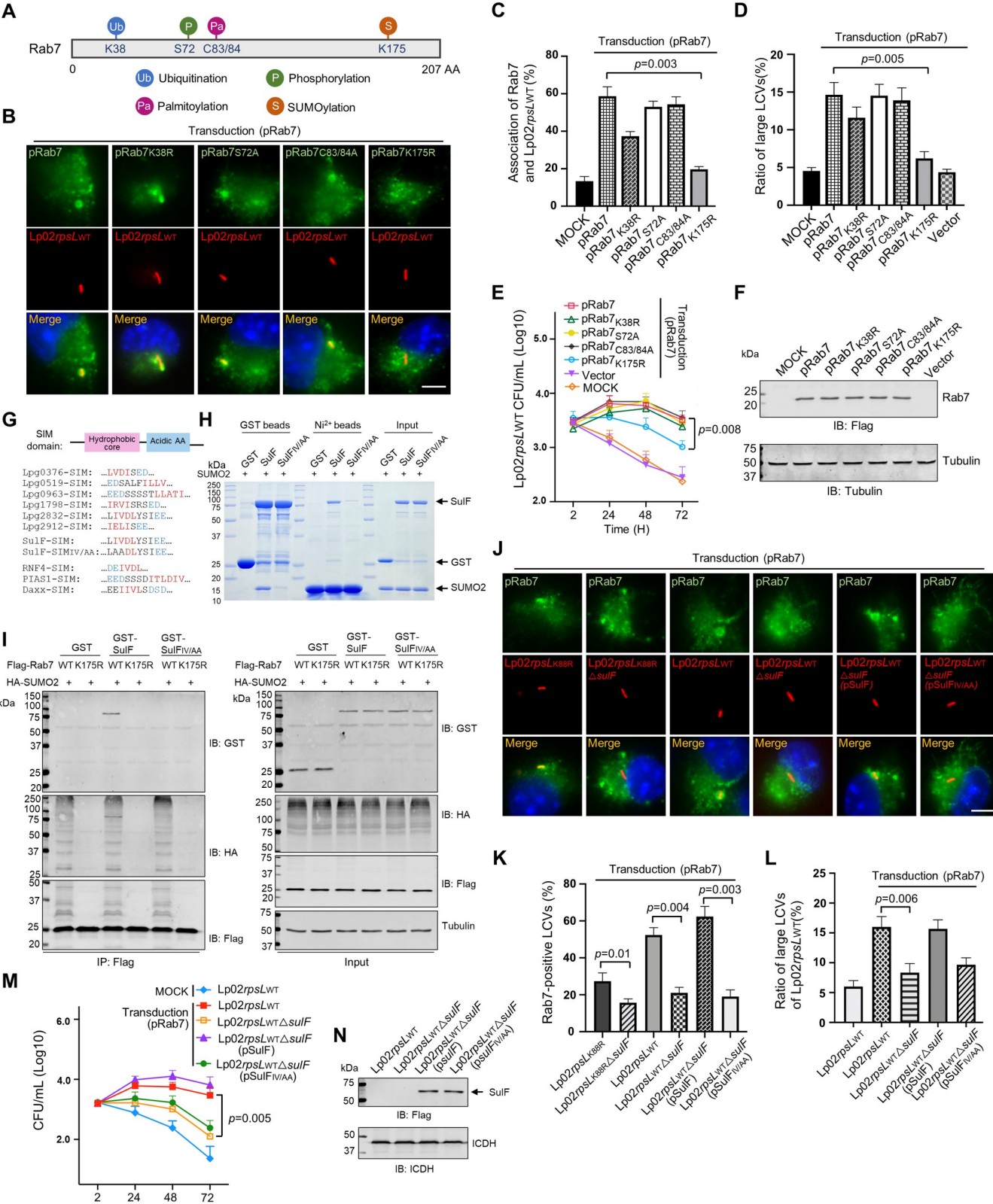

**Fig 5. SulF recruits SUMOylated Rab7 to LCVs through SUMO-SIM interaction. (A-C)** SUMOylation is required for Rab7 to be recruited to the LCV. Sketch of Rab7 with known post-translational modifications, including K38, S72, C83/84 and K175 that are modified by ubiquitination, phosphorylation,

palmitoylation and SUMOylation, respectively **(A)** Mutations in the SUMOylation site of Rab7 abolished its recruitment to the LCV. BMDMs ectopically expressing Rab7 or its mutants were infected with strain Lp02$rpsL_{WT}$, typical images of the association **(B)** and the ratios of Rab7-positive LCVs **(C)** were determined. **(D-F)** SUMOylation of Rab7 is required for its ability to support intracellular growth of strain Lp02$rpsL_{WT}$ in BMDMs. BMDMs were transduced to express Rab7 or its mutants and cells were infected with strain Lp02$rpsL_{WT}$. Bacterial growth was determined by the formation of large LCVs **(D)** and total bacterial counts **(E)**. The expression of transduced Rab7 or its alleles was probed by immunoblotting **(F)**. **(G-I)** Identification of SulF as a SUMO-binding *L. pneumophila* effector. Comparison of the sequences of the predicted SUMO-binding motifs (SIM) of Dot/Icm substrates with SIM elements in established eukaryotic SUMO-binding proteins **(G)**. The SIM motif is critical for the binding of SulF to SUMO. Recombinant GST-SulF or its SIM mutant SulF$_{IV/AA}$ and His$_6$-SUMO was mixed for 2 h and the protein complexes were captured by GST or Ni$^{2+}$ beads. After 3x washes, proteins associated with the beads resolved by SDS-PAGE were detected by CBB staining. Samples withdrawn prior to adding agarose beads were loaded as input controls **(H)**. HEK293T cells co-transfected to express HA-SUMO2 and Flag-Rab7 or its mutants for 36 h were lysed and SUMOylated Rab7 or mutants were purified. SUMOylated Rab7 and unmodified Rab7 were used to determine their binding to SulF or SulF$_{IV/AA}$ by affinity pulldown. Tubulin was probed as a loading control **(I)**. **(J-L)** Recruitment of Rab7 to LCVs by *L. pneumophila* required SulF with an intact SIM. BMDMs transfected to express Rab7 were infected with the indicated *L. pneumophila* strains for 2 h and the association of Rab7 with the LCVs was determined by acquiring images **(J)**, quantitation **(K)** and the formation and ratio of large LCVs containing more than 10 bacteria **(L)**. **(M-N)** SulF is required for intracellular growth of strain Lp02$rpsL_{WT}$ in BMDMs overexpressing Rab7. BMDMs transduced to express Rab7 were infected with the indicated *L. pneumophila* strains, bacterial growth was analyzed by determining the total counts of bacteria at the indicated time points **(M)**. The expression of SulF and its mutant in the complementation strains was detected by immunoblotting with the Flag-specific antibody **(N)**. The metabolic enzyme ICDH was probed as a loading control.

We found that the protein level of SulF in strain Lp02$rpsL_{WT}$ was higher than that of strain Lp02$rpsL_{K88R}$ in bacteria grown in bacteriological medium at all time points examined (**Fig 6A and 6B**). In agreement with this observation, the intensity of SulF staining signals on phagosomes formed by Lp02$rpsL_{WT}$ was higher than those of the common laboratory strain Lp02$rpsL_{K88R}$, and such association became undetectable on phagosomes formed by mutants lacking *sulF* (**Fig 6C and 6D**). Importantly, complementation with SulF or its SIM-deficient mutant restored its association with the LCVs but only those expressing the wild-type gene regained the ability to recruit Rab7 (**Fig 6E and 6F**).

Consistent with its higher-level expression of SulF, strain Lp02$rpsL_{WT}$ recruited more Rab7 than Lp02$rpsL_{K88R}$ (**Fig 6G**). Furthermore, introduction of a plasmid expressing SulF into strain Lp02$rpsL_{K88R}$ allowed its phagosomes to recruit more Rab7 (**Fig 6H–6K**). Yet, when examined for intracellular growth, strain Lp02$rpsL_{WT}$(pSulF) cannot productively replicate in BMDMs (**Fig 6L and 6M**), suggesting that other factors are required for strain Lp02$rpsL_{WT}$ to overcome the restriction imposed by these cells. Overexpression of SulF did not detectably impact the intracellular growth of stain Lp02$rpsL_{K88R}$ in BMDMs (**Fig 6K–6M**). Nevertheless, these results indicate that SulF mediates the recruitment of SUMOylated Rab7 to LCVs via interaction between its SIM and SUMO (**Fig 6N**).

## SulF inhibits the GTPase activity of Rab7 by blocking its interaction with a GAP

Next, we set out to determine the impact of SulF on the GTPase activity of Rab7. SUMO-Rab7 purified from mammalian cells was used for GTP hydrolysis assays (**Figs 7A and S7A**). Whereas SulF did not affect the intrinsic GTP hydrolysis of Rab7 (**S7B and S7C Fig**), it significantly inhibited the GTPase activity induced by TBC1D15, a GAP for Rab7 [64] (**Fig 7B and 7C**). The inhibition of SulF on TBC1D15-mediated GTPase activity of Rab7 occurred in a dose-dependent manner (**Fig 7D–7G**). To further explore the underlying mechanism, we examined how SulF impacts the interaction between TBC1D15 and Rab7. Although SulF did not detectably interact with TBC1D15 (**S7D Fig**), it effectively interfered with the binding of SUMOylated Rab7 to TBC1D15 (**Fig 7H and 7I**) and such interference was more apparent in assays using the GTP-bound form of SUMO-Rab7 (**Fig 7J and 7K**). In contrast, the interaction between SulF and SUMO-Rab7 was not affected by TBC1D15 (**Fig 7L and 7M**). These findings suggest that SulF recruits SUMO-Rab7 to LCVs and prevents it from being accessed by the GAP TBC1D15, thus keeping the small GTPase in its GTP-bound active form to allow proper

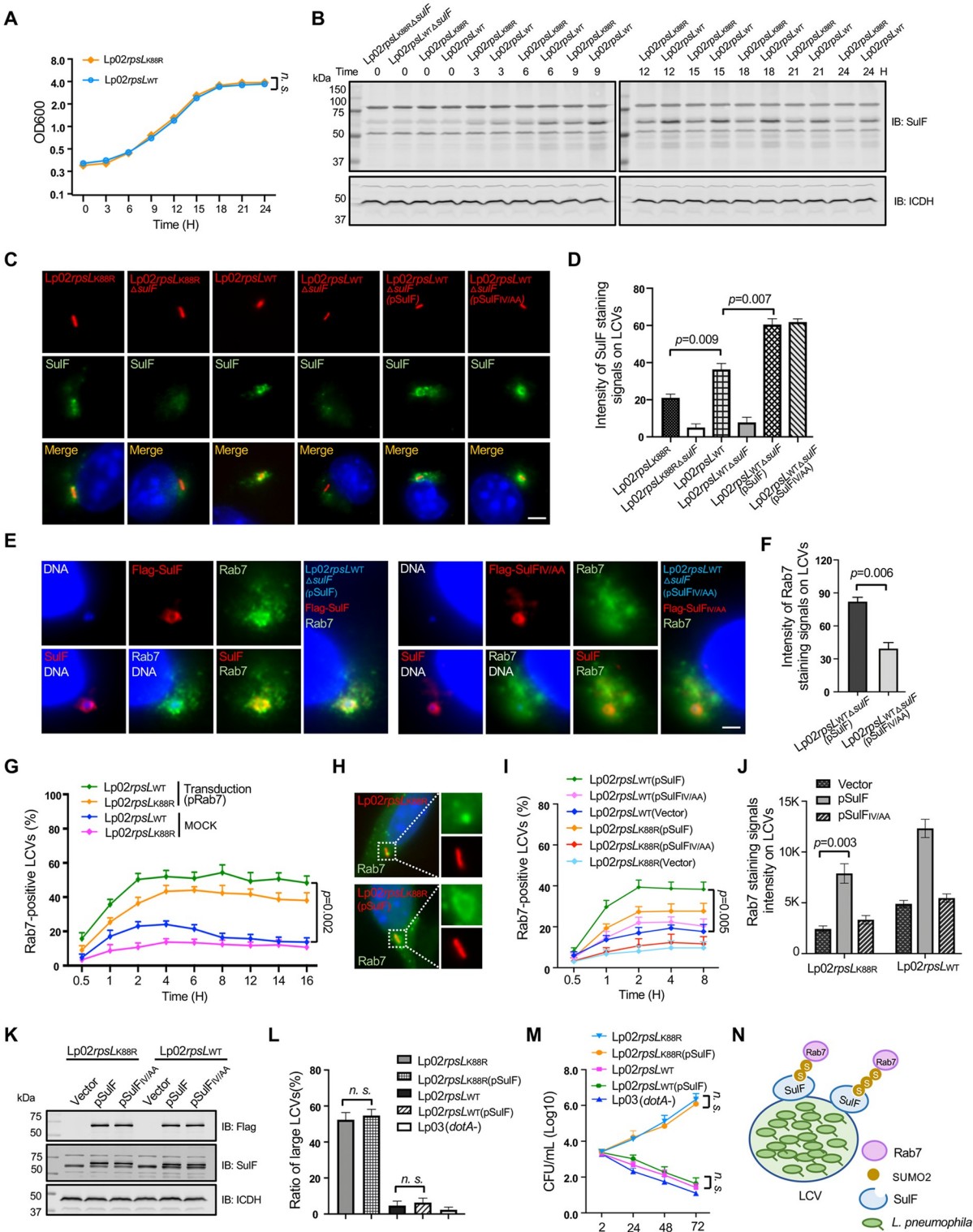

**Fig 6. Overexpression of SulF increased Rab7 recruitment by strain Lp02*rpsL*K88R.** **(A-B)** The abundancy of SulF in strain Lp02*rpsL*WT is higher than that in strain Lp02*rpsL*K88R in bacteria grown in bacteriological medium. Fresh cultures with $OD_{600}$ = 0.3 established from saturated cultures of the indicated *L. pneumophila* strains were grown in a shaker and bacterial growth was monitored by measuring $OD_{600}$ at 3-h intervals (left panel). Samples withdrawn at the indicated time points were resolved by SDS-PAGE and SulF was detected by immunoblotting. ICDH was probed as a loading control (middle and right panels). **(C-D)** The association of SulF with LCVs formed by

relevant *L. penumophila* strains. BMDMs infected with the indicated *L. penumophila* strains were processed for immunostaining with SulF-specific antibodies, images were acquired using an Olympus IX-81 microscope **(C)**. The density of SulF staining signals was measured from at least 300 phagosomes using Image J **(D)**. Bar, 5 μm. **(E-F)** The recruitment of Rab7 by *L. pneumophila* requires an intact SIM motif. Strain Lp02*rpsL*$_{WT}$△*sulF* was complemented with *sulF* or its SIM-defective mutant. BMDMs infected with the indicated bacterial strains for 2 h were processed for immunostaining and imaging **(E)**. Quantitation shown (mean ± s.e.) was from three independent experiments each done in triplicate **(F)**. Bar, 5 μm. **(G)** The kinetics of Rab7 recruitment by relevant *L. penumophila* strains. BMDMs infected with the indicated bacterial stains were sampled at the indicated time points, cells were processed for immunostaining and analyzed using a fluorescence microscope. **(H-K)** Overexpression of SulF in strain Lp02*rpsL*$_{K88R}$ increased Rab7 recruitment. BMDMs infected with the indicated *L. penumophila* strains were processed for immunostaining. Representative images used for determining the ratios of Rab7-positive LCVs **(H)** and for measuring signaling intensity **(J)**. The expression of the endogenous or plasmid-borne SulF was probed by immunoblotting with antibodies specific for Flag or SulF **(K)**. **(L-M)** Overexpression of SulF does not endow strain Lp02*rpsL*$_{WT}$ the ability to grow in BMDMs. The indicated bacterial strains were used to infect BMDMs, the distribution of LCVs **(L)** and total bacterial counts for a 72-h experimental duration at 24-h intervals **(M)** were analyzed. In all cases, quantitative results (mean ± s.e.) were from three independent experiments each. n.s., not significant. **(N)** A model for SulF-mediated recruitment of SUMOylated Rab7 by LCVs.

signaling for maintaining the integrity of the phagosome to support bacterial replication (**Figs 7N** and **S7E**).

## Discussion

Our discovery that *L. pneumophila* strains harboring chromosomal wild-type *rpsL* trigger lysosomal macrophage death in mouse BMDMs indicates that these cells encode a cryptic defense mechanism in response to such bacterial strains [26]. Using this unique experimental model and genome-wide CRISPR/Cas9 knock-out screenings, we found that *Hmg20a* is indirectly involved in restricting intracellular replication of strain Lp02*rpsL*$_{WT}$ in mouse BMDMs. Inactivation of *Hmg20a* caused an increase in the abundancy of proteins implicated in lysosome biogenesis, including LAMP1 and Rab7, which stabilize the phagosome to counteract the yet-unknown phagosome destruction mechanism to promote intracellular bacterial replication.

The development of the LCV in BMDMs starts immediately after phagocytosis and the LCV is converted into a compartment resembling the ER within minutes as it acquires proteins typical for this organelle [65]. COPII vesicles originating from the ER appear to be the main source of membrane materials required for the expansion of the LCV [66,67]. The quick conversion of the plasma membrane of nascent LCV into an ER-like compartment is believed to allow the bacterium to effectively evade the assault from the lysosomal network [8]. Consistent with this notion, a large array of Dot/Icm effectors function to modulate key regulators of membrane trafficking between the ER and *cis*-Golgi, particularly the Rab1 small GTPase [9].

Increasing evidence suggests that the LCV also acquires proteins of the lysosomal network, particularly after the initial rounds of replication [68]. For instance, at 18 h post-uptake, approximately 70% of LCVs exhibited association with LAMP1, and the pH of these vacuoles averaged at 5.6 [68]. Thus, *L. pneumophila* has the ability to survive and replicate within a compartment with lysosomal features [68]. Furthermore, the association of lysosomal proteins such as Rab7 and LAMP1 on phagosomes formed by the Philadelphia 1 strain of *L. pneumophila* was observed more than a decade ago [69,70]. Our results have provided a molecular explanation for such recruitment. The fact that SulF-mediated recruitment of Rab7 to the LCV is important for optimal intracellular replication of *L. pneumophila* indicates that the bacterium actively participates in acquiring lysosomal proteins to its phagosome. Intriguingly, SulF-dependent enrichment of Rab7 occurs within the first 2 h of bacteria uptake, suggesting that the acquisition of ER materials and the recruitment of lysosomal proteins are not mutually exclusive. Thus, it is likely that multiple protein recruitment events appear to occur concurrently, but only one exerts dominance at a particular phase during the LCV maturation process.

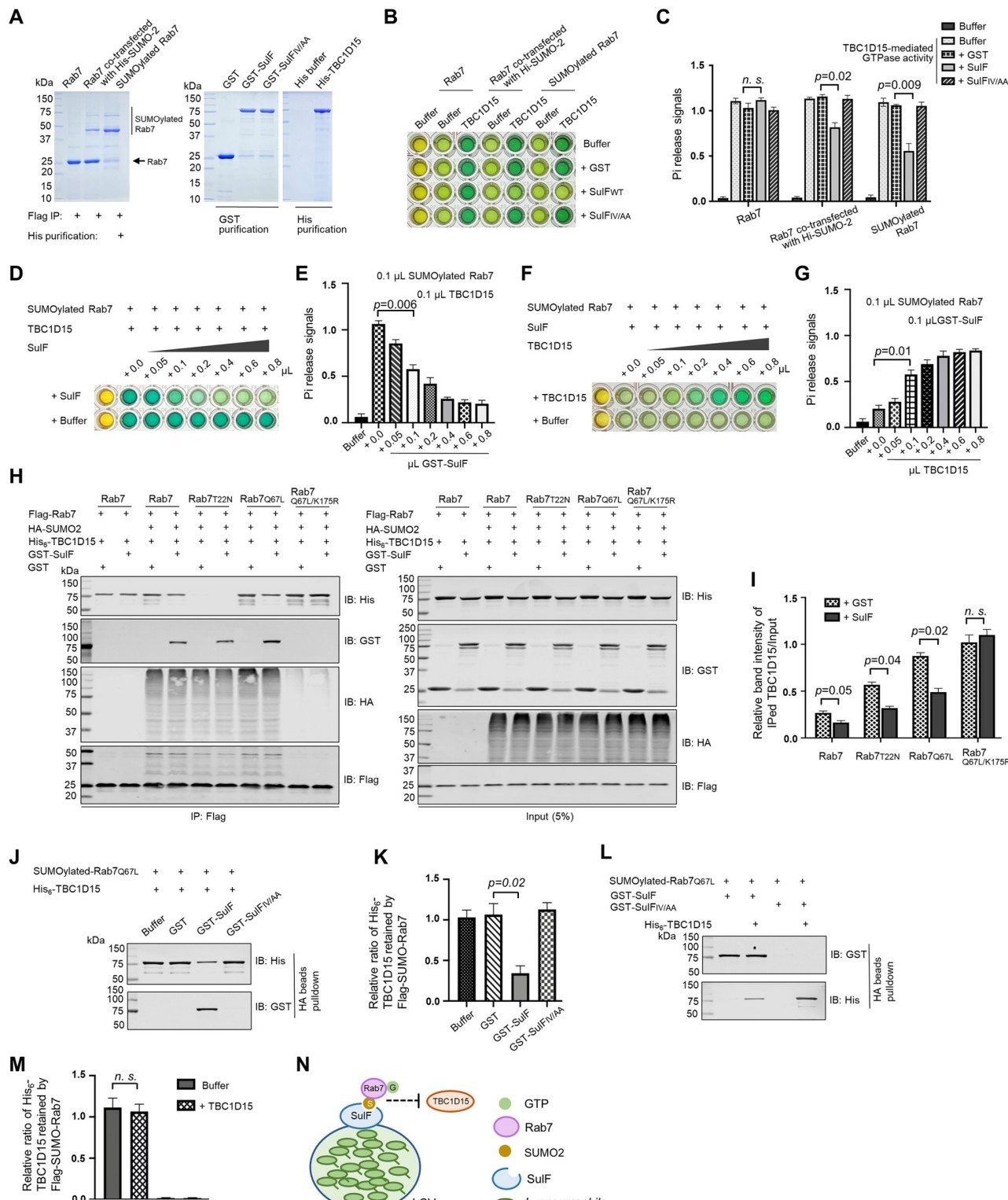

**Fig 7. SulF inhibits the GTPase activity of Rab7 by blocking the binding of a GAP protein. (A)** Purification of SUMOylated Rab7. Flag-Rab7 and His$_6$-SUMO2 were co-expressed in HEK293T cells, SUMO-Rab7 was purified by a two-step procedure using antibody specific for Flag and Ni$^{2+}$ beads. Purified proteins separated by SDS-PAGE were detected by CBB staining. **(B-C)** SulF inhibits TBC1D15-induced GTPase activity of Rab7 in an SIM-dependent manner. SUMOylated Rab7 were co-incubated with TBC1D15 and SulF or its mutant for 15 min. Then the reactions were stopped and released Pi was measured. The image of a typical plate for the assay **(B)** and quantitative results (mean ± s.e.) from three independent experiments **(C)**.

**(D-E)** SulF inhibits TBC1D15-induced GTPase activity of Rab7 in a dose-dependent manner. TBC1D15 and SUMOylated Rab7 were co-incubated with different ratios of SulF for 15 min. Then the reactions were stopped and released Pi was measured. The image of a typical plate for the assay **(D)** and quantitative results (mean ± s.e.) from three independent experiments **(E)**. **(F-G)** TBC1D15 induces GTPase activity of Rab7 in a dose-dependent manner. SulF and SUMOylated Rab7 were co-incubated with different ratios of TBC1D15 for 15 min and the reactions were stopped and released Pi was measured. The image of a typical plate for the assay **(F)** and quantitative results (mean ± s.e.) from three independent experiments **(G)**. **(H-I)** The impact of SulF on the binding of TBC1D15 to Rab7 requires SUMOylation. Lysates of cells transfected to express HA-SUMO2 and Flag-Rab7 variants were subjected to immunoprecipitation with Flag beads, and agarose beads coated with the indicated Rab7 or its mutants were used to evaluate their binding to TBC1D15 in reactions with or without SulF. Proteins retained by Flag beads were analyzed by immunoblotting with appropriate antibodies **(H)**. Relative ratio of the intensity of TBC1D15 bands in immunoprecipitation samples to those in input. Results shown (mean ± s.e.) were from three independent experiments **(I)**. **(J-M)** SulF interferes with the interactions between TBC1D15 and SUMOylated Rab7. His$_6$-TBC1D15 was added to HA beads coated with Flag-SUMO-Rab7 (HA-SUMO) prepared as described in A. After 1 h incubation, GST-SulF or GST was added. His$_6$-TBC1D15 retained by the HA beads was analyzed by immunoblotting **(J-K)**. GST-SulF or its mutant was incubated with Flag-SUMO-Rab7 prior to the addition of His$_6$-TBC1D15 for 4 h, His$_6$-TBC1D15, GST-SulF or its mutant retained by HA beads was detected by immunoblotting **(L-M)**. Results shown (mean ± s.e.) were from three independent experiments **(K and M)**. **(N)** A model for the mechanism of action of SulF. SulF is anchored on the cytoplasmic surface of the LCV where it recruits SUMO-Rab7. The binding of SulF to the small GTPase prevents it from being accessed by the GAP protein TBC1D15.

Although the bacterial factors directly responsible for the death of BMDMs infected with strain Lp02*rpsL*$_{WT}$ remain unknown, current results suggest such factors activate a pathway that compromises the membrane integrity of the LCV [26], a damage that can at least be partially alleviated by elevating the protein level of Rab7. Mutations in RpsL, including K43N and K88R that confer resistance to streptomycin are known to impair ribosome function [71]. Such impairment likely will alter the abundancy of at least some bacterial proteins. We speculate that the K88R mutation in RpsL caused a reduction in the abundancy of one or more proteins capable of inducing lysosomal cell death in mouse BMDMs, which accounts for the gain of function phenotype in growth in these cells. In addition, the putative receptors for the yet unknown triggers from the bacterium likely are absent in human cells because no cell death was observed when such cells were infected with *L. pneumophila* expressing chromosomal wild-type RpsL. Different responses to *L. pneumophila* between macrophages from human and mouse are not unprecedented. For example, whereas mouse macrophages respond robustly to flagellin of *L. pneumophila* by activating the NLRC4 inflammasome, such response does not occur in human macrophages due to the absence of NAIP5, the receptor that directly engages flagellin [16,17].

SulF is expressed at a higher level in strain Lp02*rpsL*$_{WT}$ than common laboratory strains such as Lp02*rpsL*$_{K88R}$ (**Fig 6B**), which accounts for the more robust Rab7 recruitment by its vacuoles. However, overexpression of SulF alone is insufficient to counteract the damage initiated by the as-yet-unknown factors. Nevertheless, the phenotypes associated with strain Lp02*rpsL*$_{WT}$ have enabled the discovery of a bacterial factor responsible for recruitment of an endosomal small GTPase to LCVs, indicating that components of the late endosomal network are important for the development of the *Legionella* phagosome. SUMOylation of Rab7 occurred on K175, a residue located in the far carboxyl portion of the protein, is distal to the usual binding sites for regulatory proteins and other interaction partners [72]. The binding of SulF, a protein of approximately 63 kDa likely will induce conformational changes in Rab7 or the bulky SulF protein may directly block the access by interacting proteins such as the GAP TBC1D15.

Modulation of ubiquitin signaling is critical for the intracellular life cycle of *L. pneumophila*, but little is known about the significance of SUMOylation in its virulence [73,74]. A common mechanism used by *L. pneumophila* effectors to recruit host proteins is direct binding [9]. The requirement of SUMOylation in SulF-mediated Rab7 recruitment may provide an additional layer of regulation for the process. Rab7 is exploited by *Salmonella Typhimurium* for maturation, stabilization and subcellular positioning of its phagosomes and the bacterium appears to

inhibit its SUMOylation [59]. Whether SUMOylation is involved in other aspects of LCV development awaits further investigation.

Although knockout of *Hmg20a or Nol9* allowed strain Lp02*rpsL*$_{WT}$ to productively replicate in mouse BMDMs, the relationship between these two proteins is unknown. Like Hmg20a, Nol9 is a nucleolar protein possessing polynucleotide 5'-kinase activity, which participates in ribosome biogenesis [75]. Interestingly, knockout of *Nol9* did not exhibit any detectable impact on the expression of Rab7 (**S7F Fig**), suggesting that its role is exerted by a different mechanism. Future study aiming at identification of the bacterial factors involved in LCV damage induction and the pathways participated by these host proteins will provide new insights into the development of the *Legionella* phagosome.

## Materials and methods

### Ethics statement

All animal use procedures were in strict accordance with the NIH Guide for the Care and Use of Laboratory Animals and were approved by the Purdue Animal Care and Use Committee (PACUC) (Protocol number 04–081).

### Bacterial strains and plasmids

*Legionella pneumophila* strains Lp02*rpsL*$_{K88R}$, Lp03(*dotA*$^-$) had been described [37]. Strains Lp02*rpsL*$_{WT}$ and Lp02*rpsL*$_{K43N}$ were described earlier [26]. *L. pneumophila* strains were grown on thymidine-rich charcoal-yeast extract (CYE) plates or in ACES-buffered yeast extract (AYE) broth. Strain Lp02*rpsL*$_{WT}$ (pAM239) (GFP$^+$) used in the CRISPR/Cas9 screening was generated by introducing pAM239-GFP [76] into Lp02*rpsL*$_{WT}$. The expression of GFP was induced by incubating bacterial cultures with IPTG (0.4 mM) for 6 h prior to infection. *Escherichia coli* strains were cultured on L-agar plates or L broth (LB). Antibiotics were used as follows: For *E. coli*, ampicillin (Amp, 100 μg/mL), chloramphenicol (Cm, 30 μg/mL), kanamycin (Km, 30 μg/mL). For *L. pneumophila* were chloramphenicol (5 μg/mL), kanamycin (20 μg/mL), streptomycin (Strep, 100 μg/mL).

The Mouse Genome-wide Knockout CRISPR sgRNA Library was purchased from Addgene (#67988) [5]. The plasmid pKLV2-U6gRNA5(BbsI)-PGKpuro2A [77] for delivering sgRNA against specific genes was purchased from Addgene (#67974). Another two plasmids, pMD2.G and psPAX2, needed for lentiviral packaging were gifts from Didier Trono (Addgene #12259 and #12260, respectively). The vector pZLQ-Flag was used for expressing SulF or its mutant in *L. pneumophila* [78]. The plasmids used for ectopically expressing Rab7 in BMDMs were constructed by inserting Rab7 or its mutants into pCDH-Puro vector (Addgene #167463). For expression of proteins in HEK293T cells, genes were inserted into pcDNA3.1 (Addgene #34577) or pCMV [11]. The plasmids for prokaryotic expression are pGEX-6P-1 (NovaPro #V010912) and pET-28a (Novagen # 69864). Amino acid sequences of inserted genes were verified by bi-directional sequencing analyses.

### Mice and cell culture

A/J (strain #:000646), *Cas9*$^{+/+}$ C57BL/6 (strain #:026816) and *Hmg20*$^{+/-}$ B6 (strain #:042189) mice were purchased from the Jackson laboratory (Bar Harbor, Maine). Mouse genotypes were confirmed by PCR using primers specific for *Cas9* and *Hmg20a*, respectively. Mouse bone marrow-derived macrophages were prepared from the indicated mice of 6-10-week-old with L-929 supernatant containing heat-inactivated FBS (final concentration: 20%) and antibiotics as described previously [65]. BMDMs used for *L. pneumophila* infection were cultured in

RPMI 1640 (CORNING, 10-040-CV) medium supplemented with 10% FBS, penicillin (10,000 I.U./mL), streptomycin (10,000 g/mL) and L-glutamine. Human embryonic kidney Lenti-X 293T cells (TAKARA, 632180) used for lentiviral packaging and HEK293T cells (ATCC, CRL-3216) used for mammalian expression were grown in Dulbecco's modification of Eagle's Medium (DMEM, CORNING, 10-017-CV) supplemented with 10% FBS, penicillin, strepto-mycin, glucose and L-glutamine. All mammalian cells were cultured at 37˚C in $CO_2$ (5%) incubators.

## Method details

### *L. pneumophila* infection and intracellular replication

Mouse BMDMs were seeded in 24-well plates at the density of $4\times10^5$/well the day before infec-tion. *L. pneumophila* strains used for infections were grown to early post-exponential phase (OD600 = 3.4–3.8) in AYE broth containing the appropriate antibiotics and IPTG (final con-centration: 0.4 mM). BMDMs seeded on coverslips were infected with relevant *L. pneumophila* strains at an multiplicity of infection (MOI) of 0.05 or as indicated in specific experiments. 1 h after adding the bacteria, we synchronized infections by washing the cells with warm PBS to remove extracellular bacteria. Infected macrophages were cultured in RPMI 1640 supplied with thymidine (final concentration: 0.1 mg/mL) and placed in the incubator (5% $CO_2$ at 37˚C). Cells were lysed with 0.2% saponin at the indicated time. Properly diluted cell lysates were plated on CYET plates and incubated at 37˚C for 4 d. CFU for each sample was deter-mined from triplicate infections for each strain.

For inspecting LCVs, mouse BMDMs were seeded on coverslips at the density of $2\times10^5$/well in 24-well plates. Cells were infected with indicated *L. pneumophila* strains at an MOI of 1. Extracellular bacteria were removed by 3x washes with warm PBS 1 h post infection and infected macrophages were cultured in RPMI 1640 containing 0.4mM IPTG for an additional 13 h. After fixing with paraformaldehyde (4% in PBS) at 4˚C for 1 h, samples were permeabi-lized with 0.02% Triton X-100 at 4˚C for 10 min followed by blocking in 4% goat serum at room temperature (RT) for 30 min. Cells were then strained with rabbit anti-Hmg20a antibod-ies (1:100, Proteintech) or Rab7 (1:100, Proteintech) at RT for 1 h, followed by incubation with the appropriate secondary antibodies (1:500) conjugated with the fluorescence dyes at RT for 45 min [79]. Cell nuclei were stained at RT for 10 min with Hoechst DNA dye (1:100,000; Invi-trogen). Coverslips were then mounted on glass slides with anti-fade reagent (Vector laborato-ries). The images were captured using a fluorescence microscope (Olympus IX-81).

### Genome-wide CRISPR-Cas9 Screening

The mouse sgRNA library for genome-wide CRISPR/Cas9 screening was amplified following the procedure recommended by the manufacturer [39,77]. The preparation of lentiviral parti-cle library was performed in Lenti-X 293T cells [80] seeded in 10 plates of 15-cm dishes. 15 μg of library DNA, 10 μg of psPAX2 and 10 μg of pMD2.G were co-transfected using Lipofecta-mine 2000 and P3000 (Invitrogen) for 6 h following the manufacturer's protocol. After remov-ing the transfection reagents, cells were replenished with fresh medium and incubated for an additional 42 h. Culture supernatant was collected, filtered through a 0.45-μm membrane, ali-quoted and stored at -80˚C. The tilter of the sgRNA lentiviral library was determined by RT-qPCR using the qPCR Lentivirus Titer Kit (Abcam) following the protocol provided by the manufacturer. For large-scale screening, BMDMs from $Cas9^+$ A/J mice were seeded in 6-well plates at the density of $2\times10^6$/well. A total of $2\times10^9$ cells were transduced with the lentiviral par-ticles carrying the sgRNA library at an MOI of 0.5. After incubation for 6 days, cells were infected with strain Lp02$rpsL_{WT}$ (pGFP) grown to post-exponential phase at an MOI of 1.

After incubation for 14 h, samples collected with cold PBS buffer were fixed with 4% paraformaldehyde at 4˚C. Cells containing large LCVs were isolated though a cell sorter (BD Biosciences FACS ARIA SORP) equipped with the FACS Diva 7 software, using a 200 mW 488 nm blue laser for excitation. GFP signals were detected through a 505 long pass filter and a 525/50 emission filter.

Samples obtained from cell sorting were pooled and used for total DNA extraction with the Purogene Core Kit A provided by Qiagen (cat#: 1042601). The sgRNAs were amplified from the DNA samples by a two-step PCR procedure [5]. The first PCR was performed in six tubes and the products were pooled and used as templates for the next PCR. Amplicons from the second-step PCR reactions were isolated by electrophoresis on 2% agarose gels. The DNA amplicons were sequenced using the Illumina HiSeq2500 instrument (GENEWIZ). Gene ranking was calculated by the MAGeCK RRA method [81] and the volcano plot were generated by R version 4.1.2.

## Gene knockout by CRISPR/Cas9 in BMDMs

Gene knockout using the CRISPR/Cas9 system in mouse primary macrophages was performed following a procedure reported recently [42,82]. Briefly, sgRNAs targeting the candidate genes were cloned into pKLV2-U6gRNA5 vector [5] using the restriction enzyme site *Bbs1*. Plasmids were co-transfected with pMD2.G and psPAX2 for lentiviral packaging in Lenti-X 293T cells. Progenitor cells collected from bone marrows of *Cas9*[+] A/J mice were spin-transduced with lentiviral particles at an MOI of 50 at 1,200g at 33˚C for 2 h. Then, cells were resuspended and cultured in RPMI 1640 supplemented with 10% conditioned medium of L-929 cells and antibiotics (Amp/Strep). 3 d post transduction, non-transduced cells were eliminated by incubation in medium containing puromycin (2 μg/mL) for 2 d. Cells cultured in fresh puromycin-free medium for an additional 3 d were harvested for immunoassays or seeded into 24-well plates for infections with the appropriate *L. pneumophila* strains.

## Immunoblotting

Cells for immunoblotting analyses were lysed in lysis buffer (150 mM NaCl, 50 mM Tris-HCl at pH 7.5, 1 mM DTT, 1% Triton X-100) at 4˚C. Lysates were boiled at 95˚C for 10 min after adding 6×SDS sample buffer, proteins separated by SDS-PAGE were transferred onto 0.2 μm nitrocellulose membranes (Bio-Rad, cat#1620112). After blocking with PBST buffer containing 5% non-fat milk at RT for 1 h, membranes were incubated with the indicated primary antibodies at 4˚C for 12 h. The dilution for anti-Hmg20a, anti-Rab1, anti-Rab7, anti-LAMP1, anti-Flag antibodies each was 1:1000. Washed membranes were incubated with appropriate infrared IRDye-conjuagated secondary antibodies at RT for 45 min. Signals were detected and quantitated with an Odyssey infrared imaging system (Li-Cor's Biosciences).

## Acridine orange staining

BMDMs were transduced by lentiviral particles carrying sgRNAs or constructs that allows ectopic expression of Rab7 at an MOI of 50. Transduced cells were then seeded in 24-well plates at the density of $4\times10^5$/well, 12 h later, cells were infected with relevant *L. pneumophila* strains at an MOI of 1. Extracellular bacteria were removed 1 h post infection by 3x washes with warm PBS. After incubation for an additional 8 h, medium was removed and cells were incubated in an AO solution (5 μg/mL acridine orange dye, 0.2% glacial acid in PBS) at 37˚C for 15 min. Wells filled with cold PBS buffer were imaged using a fluorescence microscope (Olympus IX-83).

## Cytotoxicity Assay

The release of LDH was measured using the CytoTox 96 Non-Radioactive Cytotoxicity Assay kit (Promega) following the manufacturer's protocol. BMDMs seeded in 24-well plates at $4x10^5$/well were infected with the relevant *L. pneumophila* strains at an MOI of 1. After incubation for 8 or 14 h, 50-μL of lysis buffer was added to each well and the plates were then centrifugated at 1,200 g for 5 min. 50-μL supernatant from each well was transferred to a new 96-well plate and incubated with 50-μL reconstituted substrate mixture at 37°C. After incubation for 30 min, the reaction was terminated by adding 50-μL stop solution. The absorbance at 490 nm was measured using a microplate reader (Biotek). The percentage of cytotoxicity was determined by the release of LDH (%) = (Experimental LDH release-Spontaneous LDH release)/(Total LDH release-Spontaneous LDH release)x100.

## TUNEL staining

BMDMs transduced with lentiviral particles carrying specific sgRNAs were seeded in 24-well plates at $4x10^5$/well, cells were infected with the relevant *L. pneumophila* strains at an MOI of 1. The infection was allowed to proceed for 8 or 14 h, cells were then fixed as described above and were stained with TUNEL using the In Situ Cell Death Detection Kit, Fluorescein (Roche) following the manufacturer's protocol. Briefly, reaction mixtures were prepared by adding the enzyme into the reaction buffer (1:10); 50-μL of the mixture was added onto each coverslip. After incubation at 37°C for 1 h, cells were then stained with Hoechst at RT for 10 min. After 3x washes with PBS, coverslips were mounted with anti-fade reagent (Vector laboratories) for visual inspection using a fluorescence microscope.

## Mass Spectrometry

Total proteins of *Hmg20a*$^{-/-}$ BMDMs and the appropriate control cells were separated by SDS-PAGE gels for 10 min at 65 V. Protein bands visualized by Coomassie brilliant blue (CBB) staining were digested with trypsin as described previously [83]. Briefly, gel slices were cut into 1 mm cubes and incubated in de-staining buffer containing 50% Acetonitrile (ACN) and 50 mM $NH_4HCO_3$ at 37°C for 1 h. After reduction and alkylation, proteins were digested by trypsin (12 ng/μL trypsin in 50 mM $NH_4HCO_3$, 10% ACN) at 37°C for 1 h.

Digested peptides were analyzed by liquid chromatography (LC)-ESI-MS/MS using the Dionex UltiMate 3000 RSLC nano System coupled to the Q ExactiveTM HF Hybrid Quadrupole-Orbitrap Mass Spectrometer (Thermo Fisher Scientific) following the procedure described previously [84]. Briefly, peptides were separated using a trap column (300 μm ID × 5 mm) packed with 5 μm 100 Å PepMap C18 medium and a reverse-phase column (50 cm long × 75 μm ID) packed with 2 μm 100 Å PepMap C18 silica. For solvent A, the mobile phases are 0.1% formic acid (FA), and for solvent B, 0.1% FA in 80% ACN. The mass spectrometer had been configured in positive ions and regular data-dependent acquisition mode, and the Advanced Peak Detection function was set to on for the top 20n. With a stepped normalized impact energy setting of 27%, the precursor ions were fragmented. Resolution was set to 120,000 for MS1 and 15,000 for MS2 on the Orbitrap mass analyzer.

MaxQuant software (version 1.6.3) was used to analyze the LC-MS/MS data by comparing it to the *Mus musculus* database (UniProt). For the searches, the following settings were evolved: precursor mass tolerance of 10 ppm; trypsin enzyme specificity allowing up to 2 missed cleavages; oxidation of methionine (M) as a variable modification and carbamido-methylation of cysteine (C) as a fixed modification. For peptide spectral matching (PSM) and protein identification, the false discovery rate (FDR) was adjusted to 0.01. LFQ intensity values were utilized to quantify the relative protein abundancy. Only proteins detected with at least

one unique peptide and MS/MS $\geq$ 2 (spectral counts) were deemed authentically identified and utilized for further research. Logarithmic values (Log$_2$) of LFQ intensity were further processed using Persus software (version 1.6.15.0), and random integers drawn from a normal distribution (width = 0.3, shift = 1.8) were used to fill in the gaps where data was absent.

## RNA isolation and Reverse Transcription qPCR

The mRNA level of the indicated host genes in *Hmg20a$^{-/-}$* samples was determined by RT-qPCR [85]. Total RNA was isolated using TRIzol following the manufacturer's protocol. Briefly, cell pellets were resuspended in 1 mL TRIzol and incubated for 3 min. Next, 200-μL chloroform was added to each sample and the mixture was incubated for 5 min. After mixing by short vortexing, samples were centrifugated at 12,000 g at 4˚C for 15 min. Then 700-μL cleared phase was transferred to a new tube and mixed with 700-μL isopropanol by shaking vigorously. After centrifugation at 4˚C for 15 min, precipitated RNA was washed with ethanol. RNA was precipitated again and resuspended in H$_2$O. RT-qPCR was performed using a One-step BrightGreen RT-qPCR Kit (Abm, G471-S) following the manufacturer's protocol in the LightCycler 96 Real-Time PCR System (Roche). Isocitrate dehydrogenase (ICDH) and glyceraldehyde 3-phosphate dehydrogenase (GAPDH) were used as a loading control for *L. pneumophila* and BMDMs, respectively. The RT-qPCR program used was 42˚C for 15 min (Reverse transcription), 95˚C for 10 min (Pre-Denaturation/Enzyme activation), 40 cycles of 95˚C for 15 sec (Denaturation) and 60˚C for 60 sec (Annealing/Extension). Sequences for all the primers are listed in Key resources table.

## Protein purification

For the expression of GST- or His$_6$-tagged recombinant proteins, 500 mL *E. coli* cultures were cultivated in LB medium supplemented with 30 μg/mL kanamycin or 100 μg/mL ampicillin at 37˚C until reaching an OD600nm of 0.8. Protein expression was induced with 0.4 mM IPTG at 18˚C for 16 h. After collecting bacterial cells by centrifugation, the pellets were lysed in TBS buffer containing 0.5% Triton and subjected to sonication (12%, 2s ON, 2s OFF). The soluble lysates obtained after two rounds of centrifugation at 18,000g and 4˚C were incubated with glutathione agarose beads (Pierce) or Ni$^{2+}$-NTA beads (QIAGEN). Recombinant proteins were eluted from the beads using 10 mM reduced glutathione in a Tris (50 mM Tris-HCl, pH 8.0) or TBS buffer containing 300 mM imidazole. The purified proteins were subsequently dialyzed in TBS buffer containing 10% glycerol, 0.5% Triton and 0.5 mM DTT at 4˚C for 8 h.

For the GST pull down assay, 10-μL glutathione agarose beads coated with GST-tagged recombinant proteins were incubated with 5 μg purified SUMO2 in 1 mL TBS buffer containing 0.5% Triton at 4˚C for 2 h. After three washes with TBS buffer, the eluted samples were separated using SDS-PAGE and subsequently stained with CBB.

## GTPase activity assay

GTPase activity was assessed by quantifying the released phosphate resulting from GTP hydrolysis as previously described [86]. SUMO-Rab7 was isolated from mammalian cells expressing Flag-Rab7 and His-SUMO2 through a two-step procedure using beads coated with the Flag antibody and Ni$^{2+}$, respectively. Purified Rab7 (either SUMOylated or unmodified) at a concentration of 0.1 μM was subjected to a 1-h incubation at room temperature in GTPase reaction buffer containing 1 mM GTP, with or without the addition of 0.1 μM TBC1D15 and 0.1 μM SulF (or the amount indicated in Figures). Following the incubation, malachite green reagent dissolved in HCl and polyvinyl alcohol was added. After 2 min, the reactions were

halted by the addition of 34% sodium citrate. The absorbance at 620 nm was then measured after a 30-min incubation.

## Quantification and statistical analysis

Statistical analyses were performed in GraphPad Prism 9. Image analyses were performed using ImageJ. Statistical details of experiments were described in the figure legends.

## Supporting information

**S1 Fig. Identification of host factors that restrict Lp02*rpsL*$_{WT}$ replication in BMDMs via CRISPR/Cas9 knockout screenings. (A)** Model of the lysosomal macrophage death triggered by infections with strain Lp02*rpsL*$_{WT}$. Ligands released by the bacterium trigger lysosomal membrane permeabilization and apoptosis in BMDMs, resulting in the termination of bacterial replication. **(B)** Generation of *Cas9*$^{+/+}$ mice with A/J background. *Cas9*$^{+/+}$ mice with B6 background were breeding with A/J mice, their off-springs carrying heterozygous *Cas9*$^{+/-}$ gene were backcrossed with A/J mice for an additional three generations. *Cas9*$^{+/+}$ mice with A/J background were generated by mating between *Cas9*$^{+/-}$ A/J mice. **(C)** Volcano plot of significant genes from samples containing large LCVs vs those containing small LCVs during CRISPR/Cas9 knockout screenings. Threshold: Fold change > 1.5, -log10 p value > 1, (*P* value <0.1). The 10 genes with the highest changes were highlighted.
(TIF)

**S2 Fig. Sorting of BMDMs containing large LCVs by flow cytometry. (A)** Forward scatter vs size scatter. Samples with estimated mouse macrophage size, complexity and infected by strain Lp02*rpsL*$_{WT}$(pGFP) were collected. Cell debris and dead cells were excluded from the sorting flow. **(B)** Size scatter heights vs weights. Single macrophages were collected. Aggregated cells were excluded from sorting flow. **(C)** Forward scatter heights vs weights. Cells stacked together were excluded from the sorting flow. **(D)** GFP signal vs background signal. BV421 was set as the background control to separate the signal away from macrophage autofluorescence. P4 represents macrophages containing large LCVs while P5 represents macrophages containing only one bacterium. **(E)** Percentage of cells in each portion. P4 or P5 accounts for 1% of total cell samples being examined. **(F)** Representative images of macrophages containing large LCVs in P4 or small LCVs in P5 sorted by flow cytometry. DNA was stained with Hoechst. Bar, 5 μm. **(G)** LCV size distribution of macrophages in P4 and P5 portions. At least 300 LCVs were scored for each sample.
(TIF)

**S3 Fig. Generation of *Hmg20a*$^{+/-}$ *Cas9*$^{+/+}$ mice in A/J background. (A)** *Hmg20a*$^{+/-}$ mice in B6 background were bred with *Cas9*$^{+/+}$ A/J mice, and their off-springs with the *Hmg20a*$^{+/-}$ *Cas9*$^{+/-}$ genotype were bred with *Cas9*$^{+/+}$ A/J mice for an additional four generations. *Hmg20a*$^{+/-}$ *Cas9*$^{+/+}$ mice with A/J background were generated by mating *Hmg20a*$^{+/-}$ *Cas9*$^{+/-}$ A/J mice with *Cas9*$^{+/+}$ A/J mice. **(B)** Intracellular replication of relevant *L. penumophila* strains in BMDMs from *Hmg20a*$^{+/-}$ *Cas9*$^{+/+}$ A/J mice. BMDMs were infected with bacteria at an MOI of 0.05. Samples were plated at indicated time points to obtain total colony-forming units. Data represents mean ± s.e. from three experiments each done in triplicate. **(C)** The distribution LCVs of different sizes formed by the indicated *L. penumophila* strains in BMDMs from *Hmg20a*$^{+/-}$ *Cas9*$^{+/+}$ A/J mice. BMDMs were infected by indicated strains at an MOI of 1 and infection were proceeded for 14 h before fixation and analysis. **(D)** Protein levels of apoptosis-associated proteins in *Hmg20a* knockout BMDMs infected by relevant *L. penumophila* strains. Cells were lysed 8 hpi and processed for immunoblotting using antibodies specific for

caspase3, cleaved RARP, Hmg20a or Tubulin. **(E)** The relative ratio of the band intensity of cleaved caspase3, cleaved RARP to Tubulin. Date shown as mean ± s.e. of relative ratio calculated from three independent experiments.
(TIF)

**S4 Fig. Overexpression of Rab7 in BMDMs suppresed cell death induced by infection with strain Lp02*rpsL*$_{WT}$. (A-C)** Cell death was measured by TUNEL staining. Images are representatives of one of the three experiments with similar results. Bar, 10 μm **(A).** Quantitative results (mean ± s.e.) were from three independent experiments each done in triplicate **(B)**. Expression of Rab7 in BMDMs were detacted by immunoblotting **(C)**. **(D-F)** Overexpression of Rab7 did not detectably impact intracellular growth of strain Lp02*rpsL*$_{K88R}$ in BMDMs. The indicated bacterial strains were used to infect BMDMs, the ratio of large LCVs **(D)** and total bacterial counts for a 72-h experimental duration at 24-h intervals **(E)** were analyzed. In all cases, quantitative results (mean ± s.e.) were from three independent experiments each. Expression of Rab7 in BMDMs were detacted by immunoblotting **(F)**.
(TIF)

**S5 Fig. Association of cellular organelles or relevant proteins with LCVs formed by Lp02*rpsL*$_{WT}$ in BMDMs. (A-D)** Representative images of LCVs associated with cellular organelles or relevant proteins in BMDMs infected by strain Lp02*rpsL*$_{WT}$ for 2 h **(A)** or 14 h **(C).** BMDMs were transduced by lentiviruses that direct Rab7 expression. 2 d post transduction, cells were infected by bacteria at an MOI of 1. Bar, 5 μm. Quantitation of the association of the indicated proteins with LCVs at 2 hpi **(B)** or 14 hpi **(D)**. At least 300 LCVs were counted for each sample. **(E)** The distribution of Rab7 in BMDMs co-infected with strains Lp02*rpsL*$_{WT}$ and Lp02*rpsL*$_{K88R}$. Note that Rab7 signals on LCVs formed by strain Lp02*rpsL*$_{WT}$ were stronger than those on strain Lp02*rpsL*$_{K88R}$. Bar, 5 μm. **(F)** Quantitation of the association of relevant proteins with LCVs in BMDMs. Results (mean ± s.e.) were collected from three independent experiments.
(TIF)

**S6 Fig. Identification of *L. pneumophila* effectors that contain a predicted SIM involved in binding SUMO and SUMOylated Rab7. (A)** Evaluation of the binding by GST pulldown assays. Lysates of *E. coli* expressing the candidate proteins were incubated with GST resins (5%) for 2 h. Washed GST resins coated with candidate effectors were incubated with purified SUMO2 for another 1 h and retained proteins were analyzed by SDS-PAGE and Coomassie brilliant blue staining. **(B)** Quantitation of the binding between candidate effectors and SUMO2. The binding affinity was calculated from the band intensity of SUMO2 to GST-effectors. The relative binding affinity was calculated by dividing the binding affinity of each GST-effector by the binding affinity of GST-SulF. Data were collected from three independent experiments and shown is the mean ± s.e. **(C)** Immunoprecipitation showing the binding of SUMOylated Rab7 and SulF. HEK293T cells were transfected to express Flag-Rab7 and HA-SUMO2 for 36 h. Beads coated with Flag-SUMO-Rab7 were incubated with recombinant effectors for 1 h and the retained proteins were detected by immunoblotting with appropriate antibodies. **(D-F)** Deletion of *sulF* from strain Lp02*rpsL*$_{K88R}$ did not impair its intracellular replication in BMDMs. The indicated bacterial strains were used to infect BMDMs, the protein level of SulF **(D)**, the ratio of large LCVs **(E)** and total bacterial counts for a 72-h experimental duration at 24-h intervals **(F)** were analyzed. In all cases, quantitative results (mean ± s.e.) were from three independent experiments each.
(TIF)

**S7 Fig. SulF does not impact the GTPase activity of Rab7. (A)** SUMO-Rab7 obtained by a two-step purification procedure was evaluated by SDS-PAGE and CBB staining. **(B-C)** SulF does not impact the GTPase activity of Rab7. Rab7 or SUMOylated Rab7 were incubated with SulF for 30 min. Then reactions were stopped and released Pi signals at 488nm were recorded using a plate reader. The image of one experiment **(B)** and the activity measured by released free phosphate from three independent experiments mean ± s.e. **(C)** were shown. **(D)** SulF did not interact with TBC1D15. GST-Rab7 or GST-SulF (or its SIM-defective mutant) was mixed with His$_6$-TBC1D15 and the potential protein complexes were captured with GST beads. Retained proteins separated by SDS-PAGE were detected by immunoblotting with the indicated antibodies. **(E)** Summary model for SulF-mediated recruitment of SUMOylated Rab7 in Hmg20a knockout macrophages. **(F)** *Nol9* knockout did not affect Rab7 expression. Proteins from the indicated cells separated by SDS-PAGE were probed with antibodies specific for Rab7 and LAMP1, respectively.
(TIF)

## Acknowledgments

We thank Dr. Lim, Seung-Oe (Purdue University) for plasmids; Dr. Uma Aryal at the Purdue Proteomics Facility for assistance in mass spectrometry; Dr. Jill Hutchcroft at the Purdue Flow Cytometry Core Facility for support in cell sorting.

## Author Contributions

**Conceptualization:** Chuang Li, Zhao-Qing Luo.

**Data curation:** Chuang Li, Jiaqi Fu.

**Formal analysis:** Chuang Li, Shuai Shao, Zhao-Qing Luo.

**Funding acquisition:** Zhao-Qing Luo.

**Investigation:** Chuang Li, Jiaqi Fu, Zhao-Qing Luo.

**Methodology:** Chuang Li, Jiaqi Fu, Zhao-Qing Luo.

**Project administration:** Zhao-Qing Luo.

**Resources:** Chuang Li, Zhao-Qing Luo.

**Supervision:** Zhao-Qing Luo.

**Validation:** Chuang Li, Jiaqi Fu, Shuai Shao, Zhao-Qing Luo.

**Visualization:** Chuang Li.

**Writing – original draft:** Chuang Li.

**Writing – review & editing:** Zhao-Qing Luo.

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
