## [Decision Letter · Decision Letter 0]

9 Jan 2024

Dear Dr. Luo,

Thank you very much for submitting your manuscript "Legionella pneumophila exploits the endo-lysosomal network for phagosome biogenesis by co-opting SUMOylated Rab7" for consideration at PLOS Pathogens. As with all papers reviewed by the journal, your manuscript was reviewed by members of the editorial board and by several independent reviewers. In light of the reviews (below this email), we would like to invite the resubmission of a significantly-revised version that takes into account the reviewers' comments.

Starting from the identification of a host cellular mechanism to restrict growth of a L. pneumophila strain, the authors present the comprehensive analyses to unveil the molecular mechanisms of LCV biogenesis which process includes interaction with the endolysosomal network in macrophages. Overall, the reviewers highly evaluated the study. However, they raised some concerns which the authors need to explain or address with additional experiments.

We cannot make any decision about publication until we have seen the revised manuscript and your response to the reviewers' comments. Your revised manuscript is also likely to be sent to reviewers for further evaluation.

Sincerely,

Tomoko Kubori, Ph.D.

Academic Editor

PLOS Pathogens

Nina Salama

Section Editor

PLOS Pathogens

Kasturi Haldar

Editor-in-Chief

PLOS Pathogens

orcid.org/0000-0001-5065-158X

Michael Malim

Editor-in-Chief

PLOS Pathogens

orcid.org/0000-0002-7699-2064

Starting from the identification of a host cellular mechanism to restrict growth of a L. pneumophila strain, the authors present the comprehensive analyses to unveil the molecular mechanisms of LCV biogenesis which process includes interaction with the endolysosomal network in macrophages. Overall, the reviewers highly evaluated the study. However, they raised some concerns which the authors need to explain or address with additional experiments.

Reviewer's Responses to Questions

**Part I - Summary**

Reviewer #1: The manuscript by Li et al. explores the significance of the human small GTPase Rab7 in the infection process by the bacterium Legionella pneumophila.

Legionella infects eukaryotic organisms, being taken up by phagocytosis of macrophages. During infection, Legionella secretes bacterial effectors that manipulate the host cell. A characteristic of this process is the formation of the Legionella-containing vacuole (LCV), where the bacteria can multiply.

In their work, the authors investigate why some Legionella strains cannot survive in bone marrow-derived macrophages (BMDM). Through genetic manipulation of BMDM, they identify a reduced presence of the human small GTPase Rab7 as the cause of decreased viability. Based on mutagenetic analysis, the authors speculate that the covalent modification of Rab7 with the protein SUMO is crucial for the successful establishment of Legionella infection. This speculation is supported by the characterization of bacterial effectors carrying specific SUMO-binding properties. One of these effectors, SulF, binds to SUMO-modified Rab7 and appears to keep it in an active state, where the enzymatically stimulated GTP hydrolysis of Rab7 is inhibited.

This work presents an astonishing discovery process, starting from an infection biology observation to the identification of a cellular mechanism and the responsible Legionella effector SulF. However, the paper largely focuses on outlining the pathway and places less emphasis on the detailed analysis of the molecular mechanisms of Rab7 manipulation by SulF. As a result, the actual knowledge gain for the Legionella infection process is limited, leading me to believe that publication in a more specialized journal would be more appropriate.

Reviewer #2: This is a comprehensive and exceptional study that identifies the molecular mechanisms by which certain strains of Legionella pneumophila (Lp) expressing a wild-type version of the ribosomal protein, RpsL, induce rapid cell death in mouse macrophages and thereby restricting Lp replication. Previously, the Luo lab found that a K88R mutation in RpsL conferring streptomycin resistance also attenuated the cell death response whereas a K43N mutation had no impact on attenuating cell death but still conferred streptomycin resistance. Here the authors performed a CRISPR screen that showed cell death was influenced by Hmg20a, a host factor that regulates lysosomal damage and an unrelated nucleolar protein Nol9 that possesses polynucleotide 5'-kinase activity. The authors go on the show that Hmg20a depletion rescues mouse macrophages from Lp induced cell death, as does Rab7 overexpression, and they discover a Lp effector, SulF that recruites SUMOylated Rab7 to the Legionella containing vacuole. Although the phenotype is restricted to mouse macrophages, this study has relevance to antibiotic resistance mechanisms and where such mutations intersect with and influence host-pathogen interactions.

Reviewer #3: The authors build upon their previous study showing that Legionella pneumophila strains harboring a WT rpsL gene induce lysosomal damage and apoptosis and cannot replicate in mouse macrophages. Through a CRISPR/Cas9 screen, they identify the host genes Hmg20a and Nol9 as being required for restricting Lp carrying WT rpsL. They focus on Hmg20a for the remainder of the manuscript and find that Hmg20a represses the expression of Rab7 and other endolysosomal proteins, and that overexpression of Rab7 rescues the ability of Lp carrying WT rpsL to replicate in macrophages. Furthermore, they find that the Dot/Icm effector SulF recruits SUMOylated Rab7 to the bacterial phagosome. Their data indicate that SulF inhibits Rab7 GTPase activity by blocking binding of the GAP TB1D15.

The studies are well executed and the manuscript tells a convoluted but interesting story. The authors’ study of the Lp02rpsLWT model enabled them to obtain new insight into the activity of a Lp effector, SulF, in recruiting Rab7 to the LCV. Furthermore, the manuscript indicates a role for Rab7 in regulating intracellular replication of Lp02rpsLWT BMDMs. Overall, their data suggest that Lp interacts with the endolysosomal network during LCV biogenesis in macrophages. However, some of the conclusions are based on findings where the authors are examining SulF overexpression or deficiency in the Lp02rpsLWT strain, often in macrophages overexpressing Rab7. I would like to see the authors perform some of the experiments in macrophages that are not overexpressing Rab7, and examine Rab7 overexpression or SulF deficiency or overexpression in the context of the Lp02rpsLK88R strain, in addition to some of my other suggestions listed below.

**Part II – Major Issues: Key Experiments Required for Acceptance**

Reviewer #1: I would like to outline my main criticisms below:

- The detailed presentation of the discovery process regarding the importance of Rab7 for infection is extensive. While it is understandable that a significant portion of the research effort is dedicated to this aspect, the exposition of the discovery process seems to contribute minimally to knowledge gain. The complexity of this detailed process significantly impacts the understanding of the work. Substantially shortening this aspect or moving it to the supplementary materials would likely enhance the overall comprehension of the study. The main focus should be on the molecular mechanisms of Rab7 manipulation by SulF.

- An important finding is the discovery that SulF appears to suppress GAP-mediated GTP hydrolysis of Rab7. However, the presented data raise concerns. Are the protein concentrations in the "GTPase activity assay" chapter (line 657) correctly stated? The concentrations of Rab7 (1mM), SulF (0.1mM), and TBC1D15 (0.1mM) seem unusually high. Could the authors possibly mean uM instead of mM? Furthermore, the inhibitory effect of SulF on Rab7 GTP hydrolysis is not explainable under the given protein concentration conditions. Since the concentration of Rab7 is 10 times higher than that of SulF and TBC1D15, 90% of Rab7 should be accessible to TBC1D15 at all times, and the GTP hydrolysis should hardly be affected. The observed effect could only be explained if SulF directly prevents the binding of the GAP by binding to TBC1D15. However, this has not been demonstrated. How do the authors explain this effect?

- The SUMOylation position (K175) of Rab7 is located far from the usual binding sites for regulatory proteins and other interaction partners. Therefore, it is difficult to conceive how the SUMO-mediated binding of SulF to Rab7 could have a competitive effect on GAB binding (TBC1D15). Even though a detailed analysis of the molecular mechanisms of SulF binding might be beyond the scope of this study, a thorough discussion of this issue would be appropriate. Could the authors speculate on whether SulF has direct or indirect effects on the action of TBC1D15?

Reviewer #2: I have no suggestions for further experiments as the study is comprehensive and large covering multiple techniques that have been performed and presented to a high standard. I have some minor suggestions to explain the purpose of the manuscript for readers who are not familiar with the original study.

1. Line 78, RpsL here needs more context of the original findings, intersection with Strep resistance and what RpsL does.

2. Please add statistical analyses to all graphs, eg. Fig 1A, 1H etc, if not significant please say so and in the figure legends, state the statistical test used.

3. The discussion should include some thoughts on why the mutant K88R does not trigger lysosomal induced cell death

4. The discussion should also compare the results with human cells, ie. why is the mechanism found in BMDM.

5. I think the intersection of antibiotic resistance mechanisms and impact on virulence could also be mentioned as this is exemplified here.

Reviewer #3: Major comments:

1. The authors show that overexpression of active Rab7 enables Lp02rpsLWT to replicate in BMDMs. The authors should test if Rab7 overexpression also leads to reduced cell death in BMDMs infected with Lp02rpsLWT, as they showed for BMDMs lacking Hmg20a.

2. Is the phenotype that Rab7 overexpression supports increased LCV formation and intracellular replication of Lp02rpsLWT specific to this strain? Or does it also extend to the Lp02rpsLK88R strain?

3. In figure 5, the authors examine large LCV formation and intracellular replication of a Lp02rpsLWT mutant lacking sulF in macrophages overexpressing Rab7, and they find that this Lp02 mutant has a defect in intracellular replication. This is an interesting finding, but it is in the context of Rab7 overexpression. The authors should also examine the phenotype of Lp02rpsLWT mutant lacking sulF as well as the Lp02rpsLK88R mutant lacking sulF in WT macrophages that are not overexpressing Rab7.

4. The authors’ data indicate that Rab7 is more robustly recruited to the vacuoles of Lp02rpsLWT than Lp02rpsLK88R, as LP02rpsLWT expressed higher levels of SulF than Lp02rpsLK88R and overexpression of SulF in Lp02rpsLK88R led to increased Rab7 recruitment to the LCV. The authors examine the ratio of large LCVs and intracellular replication in Lp02rpsLWT overexpressing SulF, but not Lp02rpsLK88R overexpressing SulF. The authors should also perform experiments to examine the phenotype of Lp02rpsLK88R overexpressing SulF in terms of large LCV formation and intracellular replication. This would help ascertain whether increased SulF expression is responsible for the Lp02rpsLWT phenotype in terms of decreased intracellular replication in WT BMDMs.

5. Based in part on the data shown in figure 7D, the authors make the conclusion that SulF interferes with the binding of SUMOylated Rab7 to TBC1D15. However, the difference in intensity of TBC1D15 bands between conditions with and without SulF is quite subtle. It would be helpful if the authors quantitated the TBC1D15 band densities and normalized to the input bands across several experiments.

**Part III – Minor Issues: Editorial and Data Presentation Modifications**

Reviewer #1: (No Response)

Reviewer #2: None

Reviewer #3: 1.The manuscript abstract’s second sentence is “The mechanism of this unique infection-induced cell death remains unknown,” which leads the reader to think that the authors will identify the mechanism in this manuscript. However, although the authors identified a role for Hmg20a and Rab7, the precise mechanism underlying Lp02rpsLWT-induced cell death is still unknown. Thus, I would suggest that the authors remove this sentence, as it is somewhat misleading.

2. Line 186: typo- “BMMDs” should be “BMDMs”

3. Line 459: typo- “C56BL/6” should be “C57BL/6”

PLOS authors have the option to publish the peer review history of their article (what does this mean?). If published, this will include your full peer review and any attached files.

Reviewer #1: No

Reviewer #2: No

Reviewer #3: No

Figure Files:

While r

---

## [Decision Letter · Decision Letter 1]

30 Apr 2024

Dear Dr. Luo,

We are pleased to inform you that your manuscript 'Legionella pneumophila exploits the endo-lysosomal network for phagosome biogenesis by co-opting SUMOylated Rab7' has been provisionally accepted for publication in PLOS Pathogens.

Best regards,

Tomoko Kubori, Ph.D.

Academic Editor

PLOS Pathogens

Nina Salama

%CORR_ED_EDITOR_ROLE%

PLOS Pathogens

Michael Malim

Editor-in-Chief

PLOS Pathogens

orcid.org/0000-0002-7699-2064

Both reviewers are fully satisfied with the revised manuscript.

Reviewer Comments (if any, and for reference):

Reviewer's Responses to Questions

**Part I - Summary**

Reviewer #1: I would like to thank the authors for their positive and purposeful response to the reviewer comments. I see no need for additional changes to the manuscript and am pleased to support publication of the paper in PLoS Pathogens.

Reviewer #3: The authors have satisfactorily addressed my concerns and suggestions.

**Part II – Major Issues: Key Experiments Required for Acceptance**

Reviewer #1: (No Response)

Reviewer #3: (No Response)

**Part III – Minor Issues: Editorial and Data Presentation Modifications**

Reviewer #1: (No Response)

Reviewer #3: (No Response)

PLOS authors have the option to publish the peer review history of their article (what does this mean?). If published, this will include your full peer review and any attached files.

Reviewer #1: No

Reviewer #3: No

---

## [Editor Report · Acceptance letter]

6 May 2024

Dear Dr. Luo,

We are delighted to inform you that your manuscript, "Legionella pneumophila exploits the endo-lysosomal network for phagosome biogenesis by co-opting SUMOylated Rab7," has been formally accepted for publication in PLOS Pathogens.

Best regards,

Michael Malim

Editor-in-Chief

PLOS Pathogens

orcid.org/0000-0002-7699-2064